**Data Availability Statement:** As per the consent forms used in this study and approved by the

# Evaluating the feasibility and acceptability of a community dialogue intervention in the prevention and control of schistosomiasis in Nampula province, Mozambique

**Sandrine Martin**[1]*, **Christian Rassi**[2], **Valdimar Antonio**[3], **Kirstie Graham**[2], **Jordana Leitão**[1], **Rebecca King**[4], **Ercilio Jive**[5]

1 Malaria Consortium, Maputo, Mozambique, 2 Malaria Consortium, London, United Kingdom, 3 Malaria Consortium, Nampula, Mozambique, 4 The Nuffield Centre for International Health & Development, University of Leeds, Leeds, United Kingdom, 5 Ercílio Jive, Direção Provincial de Saúde (Provincial Health Directorate), Nampula, Mozambique

* samartinlt@gmail.com

## Abstract

### Background

Schistosomiasis is a parasitic neglected tropical disease that ranks second only to malaria in terms of human suffering in the tropics and subtropics. Biomedical disease control interventions need to be complemented with effective prevention and health education strategies, that address the social and environmental determinants of disease. Malaria Consortium conducted an implementation research study between May 2014 and February 2016, in four districts of Nampula province, Mozambique, to test a Community Dialogue (CD) intervention to enhance schistosomiasis prevention and control. The study aimed to evaluate the acceptability and feasibility of using CD to improve communities' level of knowledge, attitudes and practices, and engagement in wider schistosomiasis prevention and control efforts.

### Methods

The feasibility and acceptability of the CD intervention was evaluated using qualitative and process evaluation data collected throughout the development and implementation phases. Qualitative data sets included key informant interviews (N = 4) with health system personnel, focus group discussions (N = 22) with Community Dialogue facilitators and participants, field observation visits (N = 11), training reports (N = 7), feedback meeting reports (N = 5), CD monitoring sheets (N = 1,458) and CD planning sheets (N = 152).

### Findings

The CD intervention was found highly acceptable and feasible, particularly well-suited to resource poor settings. Non-specialist community volunteers were able to deliver participatory CDs which resulted in increased knowledge among participants and triggered individual

ethics committees referenced in the article, the use of individual participant data for future research purposes is conditional on ethical approval for additional research questions. Data access requests will be reviewed by Malaria Consortium's Research Group, which can be contacted at research.lead@malariaconsortium.org. Interested researchers do not need membership with Malaria Consortium's Research Group to gain access to data. The Research Group will review if ethical approval is required and if it has been obtained before making the data available to the researcher.

**Funding:** The study was co-funded by a Grand Challenges in Global Health grant from the Bill & Melinda Gates Foundation (www.gatesfoundation. org, grant number OPP1098362) and, through COMDIS-HSD, by UK aid from the UK government (www.gov.uk/government/organisations/ department-for-international-development). COMDIS-HSD (comdis-hsd.leeds.ac.uk) is a Research Programme Consortium led by The Nuffield Centre for International Health & Development at the University of Leeds, UK. The funders had no role in study design, data collection and analysis, decision to publish, or preparation of the manuscript. The views expressed in this manuscript do not reflect the positions and policies of the funders. The funders had no role in study design, data collection and analysis, decision to publish, or preparation of the manuscript.

**Competing interests:** The authors have declared that no competing interests exist.

and communal actions for improved disease prevention and control. The visual flipchart was a key aid for learning; the use of participatory communication techniques allowed the correction of misconceptions and positioned correct prevention and control practices as the community recommendations, through consensus building.

## Conclusion

The Community Dialogue Approach should be embedded within neglected tropical disease control programmes and the health system to create long-lasting synergies between the community and health system for increased effectiveness. However, for behavioural change to be feasible, community engagement strategies need to be supported by improved access to treatment services, safer water and sanitation.

## Introduction

Schistosomiasis falls within the group of diseases commonly known as neglected tropical diseases (NTDs). These are diseases that disproportionally affect vulnerable people in remote and rural areas of low-income countries. Schistosomiasis is an acute and chronic parasitic disease, and ranks second only to malaria in terms of human suffering in the tropics and subtropics [1]. It affects approximately 240 million people worldwide, with up to 700 million people at risk of infection [2]. People become infected when larval forms of the parasite–released by freshwater snails–penetrate the skin during contact with infested water, such as during routine agricultural, domestic, occupational, and recreational activities. Common risk factors for infection are the lack of access to clean water and sanitation, and certain play habits of school-aged children such as swimming or fishing in infested water [3]. Although rarely lethal, schistosomiasis has a significant impact on multiple dimensions of human functioning and well-being both during childhood and later in adult life in endemic communities, impacting both physical and intellectual performance [4].

In Mozambique, one of the countries most affected by the disease, *Schistosoma haematobium*, is the main parasite species present. Countrywide, the prevalence of *S. haematobium* infection among school-age children was estimated at 47% in 2009 [5]. Nampula province, in the northern part of the country, is the worst affected area; the average prevalence among school-age children is 78%, with a number of districts recording 90% prevalence [5].

In line with the World Health Organisation (WHO) guidance [6], the Ministry of Health in Mozambique has been focusing on repeated large-scale treatment (Mass Drug Administration–MDA) with praziquantel. This is delivered by health personnel to at-risk population groups in an effort to reduce morbidity and mortality due to the infection and prevent new infections by limiting transmission. A more comprehensive approach including the provision of potable water, adequate sanitation, and snail control would also reduce transmission [7]. It is increasingly recognised that prevention strategies and the social and environmental determinants of disease need to be addressed [8]. MDA alone cannot achieve complete control as reinfection with schistosomiasis can be rapid, and, from a very few infected individuals in a community, transmission can continue through inappropriate hygiene and sanitation behaviours [9]. Lack of knowledge, negative attitudes and beliefs about schistosomiasis also contribute to poor prevention practices. Effective health education is one of the complementary interventions recommended by the World Health Assembly resolution 54.19 [6].

In Mozambique, as in many other sub-Saharan countries, health education remains a neglected component of a very resource-constrained national NTDs programme. As part of routine programming, district-level NTDs Focal Points are tasked with delivering schistosomiasis awareness sessions in primary schools; however the lack of logistics and educational resources typically limits their capacity to reach out regularly to school children in their catchment area. Also adult community members, including caregivers of children, are often excluded from these health education interventions; the few opportunities for them to learn about the disease occur during MDA programmes and interactions with health providers when seeking care for schistosomiasis' symptoms.

There is growing consensus that community engagement can play an important role in improving health outcomes. A recent umbrella review concluded that community engagement interventions can be effective in contributing to communicable diseases control in low and middle income settings [10]. There is a significant body of literature around education and engagement strategies for health issues in such settings [11–15], however few studies explore how these are applied to NTDs prevention and control.

In the context of NTDs, community engagement can play an important role: help to shape communities' understanding of NTDs and of available solutions for prevention and control; improve compliance with MDA treatment; and increase adoption of protective practices [16, 17]. However, a recurrent challenge has been for programme managers to engage communities in disease control strategies that are determined independently of these communities [18]. In rural, marginalised and resource-poor communities where NTDs most often occur, conflicts remain between community and biomedical understanding of the aetiology of disease. While engaging with communities is an essential aspect of promoting greater compliance with NTDs control interventions and fostering behaviour change, more research is needed to explore innovative and practical ways to enhance NTDs efforts through community participation [19].

In partnership with the Mozambique Ministry of Health and the Nampula Provincial Health Directorate, Malaria Consortium conducted a small-scale implementation research study to test the Community Dialogue Approach (CDA), which had shown potential for improving uptake of health services and promoting recommended behaviours in the context of community case management of childhood illnesses [20].

The study aimed to test the CDA in the context of schistosomiasis prevention and control, and to evaluate its acceptability and feasibility to improve communities' level of knowledge, attitudes and practices, and engagement in wider schistosomiasis prevention and control efforts.

The study was implemented between May 2014 and February 2016, in four districts of Nampula province. The evaluation used a mixed methods approach, including two cross-sectional household surveys conducted before and after the CD intervention to assess its impact on knowledge, attitudes and practices at population level. Results from those surveys have been published [21, 22]. In this paper, we present a detailed description of the CD intervention, and summarise findings with regard to its feasibility and acceptability, drawing on the qualitative and process evaluation data collected throughout the study.

## Materials and methods

### Intervention description

The CDA [23] is a community engagement strategy using an interactive participatory communication process of sharing information between people or groups of people aimed at reaching a common understanding and consensus to address specific issues. The approach involves

training non-specialist volunteers, referred to as Community Dialogue Facilitators (CDFs), to host regular meetings within their communities to discuss health issues.

There exist a wide range of community engagement approaches for communicable diseases each using a different set of delivery mechanisms and techniques and aiming at increasing community participation in health programmes. To situate the CDA among the range of experiences of integrating community participation or engagement into health programming, we use the modified continuum of community participation proposed by Draper et al. [11]. On this scale, ranging from mobilisation, to collaboration and then empowerment, the CDA can be located at the "lower" level of participation, described as community mobilisation, where selected community members are capacitated to conduct CDs which topics and tools are predetermined outside of the community.

The CDA described here had been previously applied in the context of integrated community case management of childhood illnesses (iCCM) in Mozambique, Zambia and Uganda [20]. It was adapted to the context of NTDs, with a particular focus on schistosomiasis, and aimed at improving communities' uptake of recommended prevention and treatment measures, such as MDA adherence, seeking help from qualified health care providers and adopting basic hygiene and sanitation protective practices. The adaptation of the approach, materials and tools to the local context was guided by a rapid qualitative assessment that explored the local aetiology of disease (signs, mild/severe forms of disease), social representation of illness (experience of the disease, possible stigma), and lifestyles of the target communities in the implementation sites.

The essential elements of the CD intervention are described in Table 1 for each implementation phase, and follow the template for intervention description and replication (TIDieR) checklist and guide [24].

**Rationale and theory.** The CD intervention is inspired by the Integrated Model of Communication for Social Change [25]. This model builds on the work of Paulo Freire, the Brazilian educator who conceived of communication as dialogue and participation [26], and describes an iterative process of information sharing which leads to mutual understanding, agreement and collective action. This CD intervention provides external stimuli in the form of community sensitisation, as well as nomination and training of volunteers, to trigger the implementation of regular community-owned dialogues about schistosomiasis prevention and control. As illustrated in the conceptual framework (Fig 1), the CD methodology prompts communities to explore the disease and how it affects them, enables participants to identify local issues and relevant solutions, and leads to collective decision on individual and communal actions to be implemented. This interactive process of both dialogue and collective decision-making is expected to address a set of behaviour determinants, such as knowledge, attitudes and norms, shift individual and collective practices, and ultimately build community ownership of issues affecting the community.

**Implementation process.** The roll-out of this CD intervention involved the following key phases:

- A community sensitisation exercise was conducted to inform local leaders and gate-keepers about the intervention and engage them in facilitating the participatory selection of non-specialist volunteers in their respective communities, following a set of criteria, to serve as CDFs. The selection of volunteers was community-led, and based on criteria similar to those recommended in the literature for community health volunteers [13]; these included residence and credibility in the community, knowledge of community norms, ability to read and write, interest in health issues, motivation and availability to engage as CDF.

**Table 1. Intervention description.**

| TITLE | Community dialogues for the prevention and control of schistosomiasis | | | | | |
|---|---|---|---|---|---|---|
| WHY | Goal: Improved understanding of the disease and increased adoption of protective behaviours. | | | | | |
| | Rationale: the Community Dialogues intervention addresses a set of constructs (determinants of behaviours) through community-owned dialogue and collective decision making to bring about individual and social change, building on community's capacity to address its own problems | | | | | |
| | Theory: see conceptual framework, inspired by the Integrative Model of Communication for Social Change (IMCSC) | | | | | |
| | **Description of intervention elements** | | | | | |
| **Intervention implementation phases** | **What (materials)** | **What (procedures)** | **Who provided** | **How** | **Where** | **When & How much** |
| Sensitization | Project fact sheet Brief guidance on CDF selection process and criteria | Community leaders representing each of the 68 communities invited by their local health centre chief | District Health Authorities, supported by the project's Research Officer | Half-day general sensitization meetings | District-level | Once at project's onset |
| Volunteers' selection | Volunteers' nomination sheet (community, name, gender, contact) | Volunteers selected by their own community and from within the community to serve as 'facilitators' | Community members under the local leader's oversight | At the discretion of each community, within a two-month period | Community level | Once. Expected 200 volunteers, with gender balance: 2 volunteers (1 male, 1 female) each in 68 communities (total 136), plus 64 additional for most populated areas or low-density areas |
| Training of Trainers | Training manual for trainers, including overall learning objectives, trainers' tips, training schedule, and describing in details each session' objectives, materials and method | Contents: Step-by step study of the training manual, group work, and role-plays | Province level NTDs coordinator with support from the project's Research officer | Trainees: district-level NTDs focal points and locally-recruited consultant trainers to be deployed to each district Duration: 5-day classroom training | At province level | Once, before training of volunteers |
| Facilitators' training | Visual flipcharts adapted to low literacy audience and covering signs and symptoms of schistosomiasis, treatment, risk behaviours, protective behaviours, MDA. Community dialogue guidebook providing basic facts on schistosomiasis, guidance on a repeatable ten-step methodology for planning, organizing and conducting dialogues, and sample discussion guides | Contents: participatory facilitation techniques, basic facts about the disease, its prevention and management, and the use of visual tools. Emphasis put on the process of a dialogue vs typical health education session: Each CD session includes three main phases: (1) Exploring how schistosomiasis affects the community; (2) then participants identify locally relevant solutions, (3) and plan for individual and communal action to address the issue. | In each district, the District NTDs focal point supported by a local consultant trainer skilled in community mobilisation and participatory approaches | Trainees: volunteers Duration: 3-day classroom training using adults-learning technique and delivered in local language | At district level | Once, before CDs implementation; 2 training sessions per district of 25 participants each. District level trainings run simultaneously across 4 districts to allow for all CDs to start at the same time |

*(Continued)*

**Table 1.** (Continued)

| | | | | | |
|---|---|---|---|---|---|
| Community Dialogues | Branded T-shirt and cap for facilitators' identification Visual flipchart Monitoring form to record data on dialogues held; Planning sheet for the community to document decisions made and keep track of implementation progress | Ten-step methodology: BEFORE 1. Study materials 2. Link up with community structure to plan together 3. Spread information to community DURING 4. Introduce the topic 5. Explore the topic 6. Identify issues and actions 7. Make decisions 8. Summarize take-aways 9. Thank participants AFTER 10. Fill in the monitoring form | Volunteers trained as 'facilitators', without external support | CDs are open to any interested community member, but not expected for all to attend; while it is expected that key community figures attend regularly the dialogues, it is not a requirement for participants on 1 dialogue to attend the following ones. | At community level, each facilitator in their vicinity | Expected average of 6 CDs per facilitator in each six-month cycle, meant to coincide with biannual MDA delivery. The intervention does not include implementation of MDA or any other provision of resources for disease prevention or control. |
| Remote support to facilitators | Mobile phone credit for each CDF | Phone credit recharge handed to each facilitator | Project's Research officer | Enable facilitators to have individual communication with district level authorities and receive support as needed (technical questions, challenges) | N/A | At the start of each CD 6-month cycle |
| Face-to-face support to facilitators | Guidelines (agenda, objectives, participants, responsibilities, methodology, group exercises and key discussion points) | Feedback meeting; All trained CDFs invited to attend | District Health Authorities, supported by the project's Research Officer | Collective participatory discussion of challenges and best practices, clarification of technical questions relating to schistosomiasis prevention and control | At district level | After about three months of each cycle of implementation of CDs |
| Onsite supportive supervision of facilitators | Observation visit detailed guide | On site Observation Visit of CD session conducted by facilitator | District health authorities, with support from the project's Research officer or Province NTDs coordinator | Monitor the CD proceedings, provide tailored support to facilitator, collect feedback from facilitator, participants and community structures | At community level, to a sample of communities selected using convenience criteria | Expected at least two observation visits per month during each CD cycle |
| **Tailoring** | The intervention model was meant to be flexible for local adaptations: CDFs were given broad guidance, but no set targets neither on number of participants nor frequency of the dialogues, nor total number of CDs to be conducted. Also the CD approach is not prescriptive allowing communities to identify their own issues and course of actions. | | | | |

- The selected volunteers then received a three-day classroom training on the basics of schistosomiasis aetiology, participatory facilitation techniques, the use of visual materials, and on basic recording and monitoring tools. The CD training manual can be founds in S1 Appendix.

- The volunteers were equipped with a set of simple tools that form the primary materials of the skills-based training, which contents and format are described in Table 1: (i) a visual flipchart adapted to low literacy audience and designed to match audience's primary information needs covering: the main signs and symptoms of schistosomiasis, its treatment, risk behaviours, protective behaviours, and basic facts on MDA. The flipchart is found in S2 Appendix; (ii) a guidebook that includes: key facts about schistosomiasis; sample discussion guides; and information outlining the repeatable ten-step methodology that serves as a

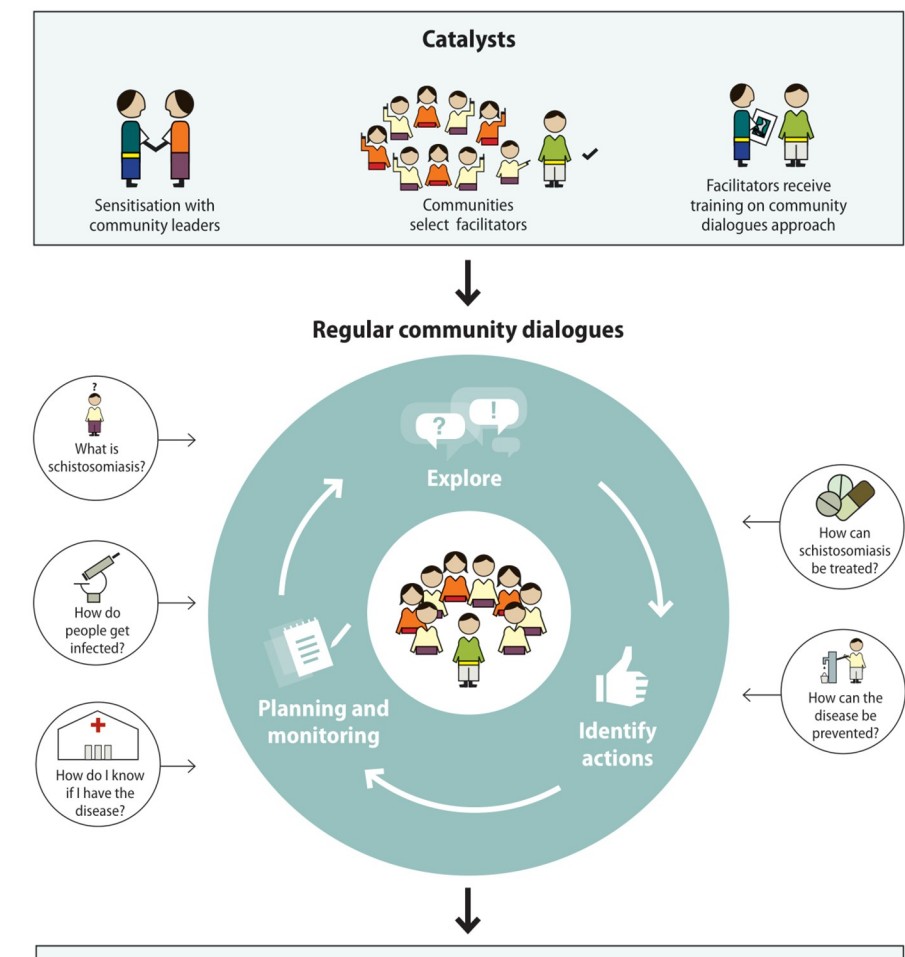

**Fig 1. Conceptual framework: CD for prevention and control of NTDs in Mozambique.**

checklist for planning, organising, conducting and documenting CDs; the guidebook is included in S3 Appendix. Trained CDFs also received (iii) identification material (branded t-shirt and cap), and (iv) a set of simple forms to record data on CDs held (monitoring form) and to document decisions made and keep track of implementation progress (planning sheet).

- Following training, CDFs were tasked with organising and autonomously delivering regular Dialogues in their vicinity, at least once per month. These were based on a suggested schedule of topics to be covered over two 6-month cycles, and meant to coincide with biannual MDA delivery.

CDs were open to any interested community member; and while it was expected that key community figures regularly attend the dialogues, it was not a requirement for participants of one dialogue to attend the following ones.

Each CD focused on a specific topic in relation to schistosomiasis, and the discussion comprised three core phases based on Paulo Freire's critical pedagogy which postulates that dialogue should provide opportunities for critical thinking, questioning of assumptions, and developing a new vision among group participants [26, 27]: (1) **Explore:** Communities are encouraged to explore a health topic through open-ended questions and open discussion. The visual flipchart provided is designed to facilitate this discussion, fill knowledge gaps and correct misconceptions. (2) **Identify:** Participants critically reflect on positive and negative behaviours pictured in the flipchart. They are encouraged to share their own stories and experiences of how the issue affects them, as well as successful and unsuccessful coping strategies. This discussion is expected to result in shaping and modelling of acceptable or desirable behaviours. (3) **Decision making:** Participants review actions and behaviours that have been identified as desirable, discussing how they could be applied in the local context, using the simple planning sheet to identify the specifics for putting decisions into action. This collective and public decision-making process is expected to result in positioning of locally relevant recommended behaviours as the social norm and facilitate planning for communal action.

CDFs were linked with the formal health system through a mobile phone credit allowance, that they were encouraged to use for contacting the district-level NTDs Focal Point, who acted as their supervisor. They could also ask questions at district-level feedback meetings, where all facilitators convened to reflect on experiences and share best practices mid-way through each CD cycle. Finally, they were linked via on-site observation visits carried out by district-level supervisors to a random sample of communities.

The intervention did not include implementation of MDA or any other provision of resources for disease prevention or control.

**Comparison with similar approaches.** In adapting the CDA, which was originally designed for the iCCM programme, three main changes were made: (1) CDFs were regular community members with a minimum literacy level, but without prior medical training or ability to provide any medical service. This is contrary to the iCCM context where dialogues are facilitated by trained community health workers (CHWs), already enrolled in an established community health programme, and who provide basic child health services (counselling, diagnosis and treatment of diarrhoea, malaria and pneumonia in children under-five). (2) CDFs were provided with a planning sheet, used during the dialogues (at decision-making stage), for participants to record decisions made, review progress and devise a course of action, whereas in the iCCM-CD model decision tracking was not documented. Instead CHWs and community leaders were expected to monitor jointly the implementation of actions agreed upon. (3) The educational content was different, given that each approach addressed a different medical subject. For the schistosomiasis dialogues, the visuals and messages were carefully

designed in close consultation with the Provincial Health Directorate, and field tested, to translate a complex disease transmission cycle into a series of risky versus protective behaviours that resonated with peoples' lifestyles, and the description of signs and symptoms most relevant to those experienced locally.

This CDA differs from other CD models previously trialled and published in Mozambique. Specifically, it differs from the *Tchova Tchova* community dialogue programme implemented for HIV prevention in Mozambique (2009–2010, provinces Zambezia and Sofala), which consisted of a series of structured sessions with pre-identified community groups, aiming at changing some underlying structural factors of HIV prevention such as gender and sexual norms [28]. In the *Tchova Tchova* intervention, CDFs received monthly monetary incentives, close mentorship and supervision, and set targets in terms of number of dialogues per topic, and number and types of participants to be reached. Instead, in the intervention described here, CDFs did not receive any monetary incentive, conducted the dialogues autonomously, and were given broad guidance, but no set targets, to allow for local adaptations. The similarities and differences of this CDA with other participatory learning and action approaches has been described elsewhere [20].

**Implementation setting.** The Nampula Provincial Health Directorate was involved in all stages of project implementation, working closely with NTDs Focal Points of the District Health Service in each of the four implementation districts. Malaria Consortium's Research Officer managed the various data collection activities and provided administrative support to the coordination and implementation of activities. The project team also engaged regularly with the national Ministry of Health NTDs Programme throughout the study. Study results were presented and discussed at a national meeting held in Maputo in November 2016, co–hosted by the NTDs Programme and Malaria Consortium, which brought together a range of stakeholders with an interest in NTD programming and those with interest in community mobilisation and health promotion, including key Ministry of Health departments, major donors, non–government organisations, UN agencies and local research groups.

CDs were implemented in two six-month cycles, between August 2014 and November 2015, and aimed to cover all 68 communities of the four targeted districts, as listed in the national census 2007 [21]. All study districts have been receiving MDA for schistosomiasis, lymphatic filariasis and soil transmitted helminths since 2009, although inconsistently due to both drug shortages and operational challenges. Schistosomiasis campaign target groups varied depending on drug availability and implementation strategies. MDAs generally targeted school-age children between five and 14 years but, during the intervention, two districts received MDA that targeted the entire population over five years of age. Beside MDAs, no other programme targeting schistosomiasis prevention and control was implemented in the study area during the study period, according to the Nampula Provincial Health Directorate.

## Evaluation methods

**Study design.** The study was conceived as pragmatic research with the objective of contributing to the development of practical recommendations for health policy and practice [29, 30]. It was designed as a small-scale pilot, focusing on determining feasibility and acceptability of a community engagement intervention to improve communities' knowledge, attitudes and practices, and engagement in wider schistosomiasis prevention and control.

A detailed evaluation plan was developed during the early stages of the study, which specified research questions and identified data sources, time and mode of data collection for each research question. Because of the focus on understanding feasibility and acceptability of the intervention, the majority of the data collected during and after implementation was

qualitative. The intervention's impact on knowledge, attitudes and practices (KAP) at population level was measured through quantitative baseline and endline cross-sectional surveys [21, 22]. Further insights were gained through triangulation with routine monitoring and evaluation data gathered by participants and the study team throughout the development and implementation phases of the study.

The following guidelines and frameworks informed conceptualisation of the study and reporting of study results:

- Feasibility of the intervention was conceived along the lines of the UK Medical Research Council's guidance on process evaluation of complex interventions [31].

  To assess the intervention's feasibility, five aspects of its actual implementation were explored: fidelity (whether the intervention was delivered as intended), reach (whether the intended audience comes into contact with the intervention, and how), dose delivered (the quantity of intervention implemented), adaptation and mechanisms of impact (how does the delivered intervention produce change?).

- We used Peters et al.'s definition of acceptability [32] which, in the context of health interventions, refers to the degree of responsiveness of the intervention to the social and cultural expectations of individuals and communities.

  The acceptability of the intervention was assessed by qualitative inquiry into participants' satisfaction with the various elements of the intervention, and into their perception of the relevance and applicability of protective behaviours promoted through the intervention, as these are influenced by their social and cultural expectations.

- Reporting of qualitative data is informed by the consolidated criteria for reporting qualitative research (COREQ) [33].

**Study area.**   The CD intervention was implemented in four districts of Nampula province: Eráti, Mecubúri, Mogovolas and Murrupula (Fig 2), with a combined total population of 839,000 [34]. Study districts were selected in consultation with the Provincial Health Directorate based on the following criteria:

- High prevalence of schistosomiasis;

- Comparable socio-geographic conditions putting the population at risk of schistosomiasis infection, with a majority practicing subsistence agriculture in or near riverbeds;

- Comparable challenges with regard to schistosomiasis prevention and control, including a lack of access to clean water.

**Sources of data.**   Two cycles of CD were implemented. The study timeline and implementation schedule is summarised in Fig 3 below.

Data sources comprise of primary qualitative data, collected after the first CD cycle (midterm) and after the second CD cycle (endline), complemented with routine monitoring data. Table 2 provides an overview of qualitative and monitoring data sources.

A description of data collection and analysis procedures is provided below.

## Primary qualitative data

**a) Data collection.**
**Focus group discussions**

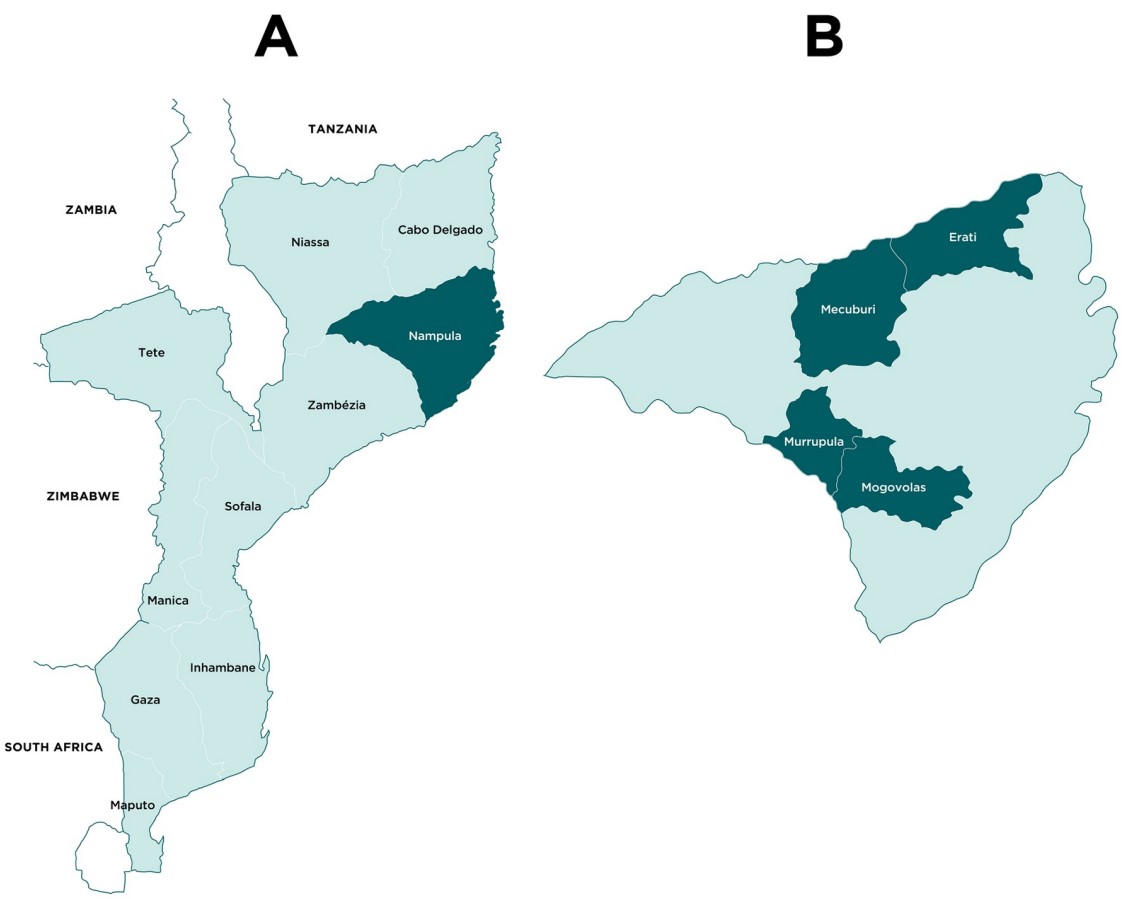

**Fig 2. Maps of Mozambique (A) and Nampula province (B) with intervention districts highlighted dark green.**

Focus group discussions (FGDs) and key-informant interviews (KIIs) were conducted at two points:

- February and March 2015, after approximately six months of implementation (midterm)

- January 2016, after completion of the two six-month cycles of CDs, conducted between August 2014 and November 2015 (endline)

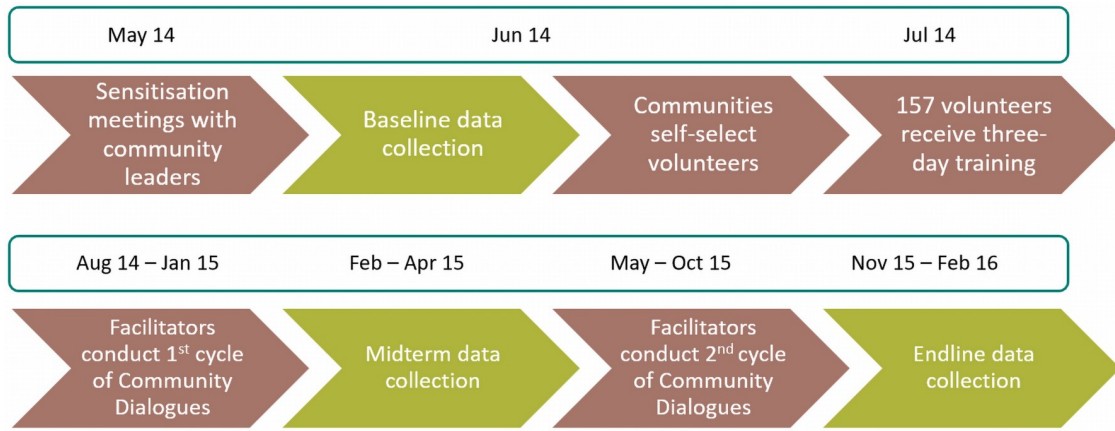

**Fig 3. Study implementation and data collection timeline.**

**Table 2. Data sources.**

| Primary qualitative data sources for thematic analysis | | Midterm | Endline | Total |
|---|---|---|---|---|
| Key-informant interviews | | - | 4 | 4 |
| Observation visits | | 11 | | 11 |
| Focus Group Discussion (FGD) | | 14 | 8 | 22 |
| FGD by target audience | sensitisation meeting attendees | 2 | - | 2 |
| | CDFs | 4 | 4 | 8 |
| | male CD participants | 4 | 4 | 12 |
| | female CD participants | 4 | | |
| Routine monitoring data sources for content analysis | | | | |
| Training reports | | 7 (1 Training of trainers report and 6 district-level training reports) | | |
| Feedback meeting reports | | 5 | | |
| CD monitoring sheets | | 1,458 | | |
| CD planning sheets | | 152 | | |

At midterm, the study team aimed to conduct one FGD each in the four intervention districts with each of the following target audiences:

- Community members who had attended the sensitisation meetings

- CD facilitators

- Female CD participants

- Male CD participants

Participants were identified using a combination of purposive and convenience sampling strategies. As a first step, suitable communities were selected using the following sampling criteria designed to elicit rich, insightful data:

- Communities that had not participated in feedback meetings

- Communities that had documented higher or lower than average number of CDs, CD participants, decisions or actions taken.

Between seven and ten communities were identified in each district. The study team subsequently invited a total of ten to twelve individuals per planned FGD from the identified communities based on convenience and in consultation with district health staff and CDFs.

Because at midterm data collection was hindered by severe flooding, FGDs with sensitisation meeting attendees could only be conducted in two of the four districts. Hence, a total of fourteen FGDs were conducted out of the sixteen planned.

At endline, the study team conducted one FGD each in the four intervention districts each with the following target audiences:

- CDFs

- CD participants

Conducting separate FGDs for male and female participants was not considered necessary as midterm responses did not vary by sex and, during the CDs, women were generally observed to be open and outspoken despite the presence of men.

Participants were sampled using a combination of purposive and convenience sampling, similar to the approach described above for midterm data collection. Between eight and ten

communities per district were selected using this method, and between ten and twelve individuals were subsequently invited to participate in each FGD, in consultation with district health staff and CDFs. A total of eight FGDs were conducted at endline.

All FGDs were facilitated by the Research Officer who was a native speaker of Macua, the language commonly spoken in the study districts. Semi-structured discussion guides were developed in English for each target group and subsequently translated into Portuguese. Discussion guides were not translated into Macua, as there is no tradition of reading and writing in this language; Portuguese being the general language of education, adults who are literate most often do not read and write easily in their local language. The Research Officer also felt more comfortable translating directly from the Portuguese versions of the guides. While it was not possible to pre-test the discussion guides, the team discussed challenges and required adaptations after the first round of FGDs with each target audience.

All FGDs were audio-recorded and lasted between 60 and 90 minutes. A temporary Research Assistant, a university graduate and native speaker of Macua, supported note-taking during the FGDs and subsequently transcribed the FGDs, including non-verbal clues, translating verbatim into Portuguese directly from the audio recording. The Research Officer was responsible for supervising the Research Assistant and providing quality assurance by checking the accuracy of the translation and transcription. Quotes originally in Portuguese in the data set have been translated into English for this publication.

### Key informant interviews

At endline, KIIs were conducted with a district health official with responsibility for NTDs prevention and control in each of the four intervention districts to explore their perceptions of the intervention. Interviewees were identified in consultation with the provincial NTDs Coordinator. In three districts, the interviewee was the district-level NTDs Focal Point. In one district, the NTDs Focal Point was not available and a Medical Officer was interviewed instead. All interviews were conducted by the Research Officer in Portuguese, using a semi-structured question guide, with a Research Assistant taking notes. It was not possible to pre-test the guide. Interviews were audio-recorded and lasted between 30 and 60 minutes. They were transcribed verbatim by the Research Assistant.

### Observation visits

A sample of CDs were observed and documented by the Research Officer throughout the intervention period. The Research Officer aimed to visit approximately equal numbers of communities in all four intervention districts and to observe one or two CDs each month. The Research Officer randomly contacted CD facilitators to enquire about plans to conduct CDs in the near future. He was typically accompanied by the provincial and the respective district NTDs Focal Point. Conducting the planned number of observation visits proved challenging, as facilitators tended to schedule CDs only a few days in advance, while planning field visits required up to a week's notice to ensure availability of transport to travel to the communities, as well as availability of provincial and district health staff to accompany the Research Officer to the field. In addition, planned CDs were also occasionally postponed by facilitators at short notice to accommodate the changing priorities of the community, for example for presidential or government visits, funerals or religious ceremonies. As a consequence, only eleven observation visits were conducted over the two six-month CD cycles.

During the observation visit, the Research Officer did not interfere in the CD proceedings and, with the help of an observation checklist, took notes, including direct quotes, relating to how the dialogues were conducted, topics discussed and decisions made. After the CD, the Research Officer and district health representative spent an additional one to two hours in the community to interact and have informal discussions with the CDF and two to three participants about their perceptions of the dialogue and their general thoughts on the approach. An

unstructured discussion guide was developed for this purpose. While the CDs and informal discussions were typically conducted in Macua, the Research Officer submitted a report written in Portuguese after each visit using a standardised template, summarising his observations and providing a subjective assessment of the proceedings. Direct quotes were translated into Portuguese by the Research Officer.

**b) Qualitative data analysis.**

Observation visit reports, FGD and KII transcripts were managed and thematically analysed using MAXQDA software [version 12] (VERBI GmbH). The coding frame provided in S1 Table was initially developed by CR, KG and SM based on a review of the literature and the team's experience of implementing similar interventions. It was then applied to the data set by JL, and further augmented and modified taking into account emerging themes. Modifications to the initial coding frame were discussed between JL and CR until consensus was reached, referring to the data set as required. Themes were analysed by code and summarised in a comprehensive report by JL. Taking into account limitations with regard to budget and time, the study team was satisfied that quantity and quality of the qualitative data collected was sufficient to answer research questions in sufficient depth in order to develop pragmatic policy-oriented recommendations.

## Monitoring and evaluation data

**a) Data collection.**

**Intervention development report**

A report summarising the development of the CD intervention, including rapid qualitative assessment conducted to inform adaptation of the approach to a schistosomiasis context was compiled by SM.

**Sensitisation report**

The Research Officer who conducted the sensitisation meetings with community representatives compiled a report, which included names and contact details of meeting attendees and summarised information shared.

**Training reports**

A total of seven training reports were included. A report summarising the training of trainers was compiled and submitted by the trainer. It included the training agenda, discussion of training objectives and an overall assessment of trainers' capacity. The six subsequent training sessions for CD facilitators were each documented by trainers using a report template provided by the study team. The reports included a list of all candidates trained, agenda, objectives, compiled pre- and post- training test results, and trainers' observations with regard to strengths and weaknesses of the training. Trainees' satisfaction was assessed with the help of a simple multiple-choice feedback questionnaire.

**Feedback meeting reports**

Five feedback meetings with CDF (one for each district from cycle 1 and one summary report from cycle 2) were documented by the Research Officer, who conducted the meetings using report templates provided by the study team. The reports captured the agenda, objectives, feedback received, challenges encountered and information provided to CDFs.

**Routine monitoring and evaluation**

Two simple forms were included in the CDF toolkit to collect routine monitoring and evaluation (M&E) data throughout the intervention period:

• A feedback form that CDFs were asked to complete after every CD they conducted. The form captured basic information such as time and date of the dialogue, numbers of

participants and topics covered. Facilitators were also asked to document technical questions they were unable to answer and any challenges encountered before or during the dialogue.

- A planning sheet that facilitators were instructed to complete jointly with participants to capture decisions made during the Dialogue. The form was designed as a working document to be used and revisited throughout a CD cycle. Facilitators were also encouraged to use the forms for action planning and monitoring.

The routine M&E forms were collected at feedback meetings. A total of 1,458 feedback forms and 152 planning sheets were collected over the course of the intervention. CDFs reported that they occasionally forgot to complete forms after dialogues, lost forms they had completed or neglected to submit them to the study team.

**b) Analysis of routine monitoring and evaluation data.**

SM and CR conducted a review of reports (intervention development, sensitisation, training, feedback meetings reports) received throughout the intervention and extracted key information into summaries relevant to the evaluation of the CD intervention.

Routine M&E forms (planning sheet and feedback form) were reviewed and summarised by VA with the help of a spreadsheet template capturing key evaluation themes relating to feasibility and acceptability. Summaries were further reviewed and analysed by SM and CR.

## Ethics

Ethical approval for the study was granted by the University of Leeds School of Medicine Research Ethics Committee (SoMREC/13/071) and the Comité Nacional de Bioética para Saúde in Mozambique (42/CNBS/2014). Participation in all intervention activities was voluntary. Attendees of the sensitisation meeting and Community Dialogue facilitator training participants were informed that the intervention formed part of a research study. CDFs were asked to share this information with participants during the first CD they conducted. Written informed consent was received from FGD and KII participants. For CDs observed by the Research Officer, where data was directly recorded, oral consent was received from CDFs and participants at the beginning of the community visits. All research data has been anonymised.

## Results

This section summarises findings across two main domains: 1) feasibility of the implementation process (including fidelity, reach, dose delivered, adaptation and mechanisms of impact), and 2) acceptability factors, including participants' satisfaction with the intervention, and their perception of the relevance and applicability of protective behaviours promoted through the intervention.

The findings are organized by main themes and sub-themes for each domain, as summarized in Table 3 below.

### 1. Feasibility of using the community dialogue in the prevention and control of schistosomiasis

**1.1 Fidelity.** In general, the essential elements of the CD intervention described in Table 1 for each implementation phase were delivered as planned; however some variations occurred in the initial stage of the intervention (sensitisation and volunteers' selection). The process for identifying and inviting communities to the sensitisation meeting, and the resulting CDF selection process, differed from the planned intervention design.

*1.1.1 Sensitisation.* Sensitisation of target communities was overall implemented as planned whereby District Health Services invited, through the health facilities, representatives from

**Table 3. Main themes and sub-themes.**

| 1. FEASIBILITY | Main theme | Sub-theme |
|---|---|---|
| | 1.1 Fidelity (whether the intervention was delivered as intended) | 1.1.1 Sensitisation |
| | | 1.1.2 Volunteers' selection |
| | 1.2. Reach (whether the intended audience comes into contact with the intervention, and how) | 1.2.1 CDFs recruited |
| | | 1.2.2 Communities reached |
| | 1.3 Dose delivered (the quantity of intervention implemented) | 1.3.1 CDFs training |
| | | 1.3.2 CDs conducted |
| | | 1.3.3 CD attendance |
| | 1.4 Local adaptations of the intervention's implementation | 1.4.1 CDFs' catchment area |
| | | 1.4.2 Topics discussed |
| | | 1.4.3 Support to the CDFs |
| | 1.5 Mechanisms of impact (how does the delivered intervention produce change?) | 1.5.1 Ten-step methodology |
| | | 1.5.2 Exploring the health topic |
| | | 1.5.3 Identifying actions to resolve issues |
| | | 1.5.4 Decision Making on individual and communal actions to be implemented |
| | | 1.5.5 Participatory discussion |
| 2. ACCEPTABILITY | Main theme | Sub-theme |
| | 2.1 Participants' satisfaction | 2.1.1 An engaging platform |
| | | 2.1.2 Satisfaction driven by knowledge gains |
| | | 2.1.3 Appreciation of the visual flipchart |
| | 2.2 Relevance and applicability of protective behaviours promoted through the intervention | 2.2.1 Shift in care-seeking practices |
| | | 2.2.2 Hygiene and water-handling practices |
| | | 2.2.3 Practices with limited applicability |
| | | 2.2.4 Suggestions for improvement |

communities to explain the purpose of the intervention, time-line, and roles and responsibilities with a focus on engaging community leadership in facilitating the participatory selection of non-specialist volunteers in their respective communities. There was inconsistent understanding of the unit for selection of facilitators across the four districts. The study team defined the 68 villages used as enumeration areas in the most recent census [34] as units for the intervention and planned to train two CDFs per enumeration area. However, District Health Services invited, through the health facilities, representatives from communities in their catchment area as locally defined and understood, which did not match the census list of 68 communities, especially in one district (Mogovolas). The study team later realised that there was no commonly recognised definition of the term 'community' in the intervention area, which may refer to a neighbourhood, village, or cluster of villages, depending on stakeholders' understanding.

*1.1.2 Volunteers' selection.* Participants in sensitisation meetings described the first step in selecting CDFs commonly involved liaising with their respective community leader to share the information received at the sensitisation meeting and the criteria for selection of CDFs. In most cases, the process of selection of CDFs appears to have involved a type of consensus, which seemed to vary from one community to another. In some instances, the CDFs were selected by community leaders, alone or in conjunction with health professionals, or the community leader had the final word on selection after community level discussion. The extent of participation of community members in this selection process was unclear from respondents' accounts but, overall, sensitisation meeting attendees thought the method was appropriate as

**Table 4. CDFs by district, gender and position in their community.**

| District | CDFs | Men | Women | Community Leaders | Traditional Healers | Community Health Workers | Health Activists | Other Volunteers | Others |
|---|---|---|---|---|---|---|---|---|---|
| Mogovolas | 48 | 26 | 22 | 3 | 1 | 3 | 4 | 7 | 30 |
| Mecuburi | 44 | 38 | 6 | 16 | 4 | 10 | 6 | 3 | 5 |
| Murrupula | 41 | 25 | 16 | 3 | 1 | 10 | 13 | 14 | 0 |
| Erati | 24 | 15 | 9 | 12 | 2 | 1 | 3 | 1 | 5 |
| TOTAL | 157 | 104 | 53 | 34 | 8 | 24 | 26 | 25 | 40 |

they trusted community leaders and health professionals to identify the most adequate person to fulfil the CDF role.

> The selection method in fact was good and fair, because the community knows who deserves (to be a facilitator), who is who, who is going to have the time to dedicate, talk, engage and dialogue. Because it has to be a trusted person in the community. We should not be authoritarian in saying let's choose this guy. We already know who can, we know the community where we live and maybe we know colleagues in the profession. But it is the community that knows exactly, the leaders themselves and the community itself. So the selection method was excellent.

KII, endline

**1.2 Reach.** *1.2.1 CDFs recruited.* As a result of the deviation in identifying and inviting communities to the sensitisation meeting, the number of CDFs recruited was below the number initially planned. We aimed to recruit a minimum of two CDFs, one male and one female, for each of the 68 communities across the four districts (total 136 CDFs), plus an additional number of CDFs for the most populated areas or areas with geographically dispersed population. Considering the budget limitations of the study, the study team aimed for a maximum of 200 CDFs. In practice, a total of 157 CDFs (24 in Erati, 44 in Mecuburi, 48 in Mogovolas, and 41 in Murrupula) were enrolled in the project, 53 females and 104 males. With the exception of the higher recruitment of males, the set of selection criteria provided, and described in Table 1, was broadly followed. Most CDFs were community members who had been previously involved in some health programme or activity, or community leaders as detailed in Table 4.

*1.2.2 Communities reached.* Only 40 of the villages used as enumeration areas in the census had at least one CDF enrolled at the end of the sensitisation and selection exercise as shown in Table 5. The gap in communities covered is mainly attributable to an issue associated with the district of Mogovolas, and more precisely to the Nametil locality, where a large number of census villages were not covered. It was not possible to reconcile the list of census villages with the list of communities used at health centre level, due to varied local understandings of the term

**Table 5. Communities targeted vs. reached by the intervention.**

| District | No. of census villages | No. of CDFs enrolled | No. of census villages with at least 1 CDF | % of communities with at least 1 CDF |
|---|---|---|---|---|
| Erati | 8 | 24 | 8 | 100% |
| Mecuburi | 12 | 44 | 11 | 92% |
| Mogovolas | 35 | 48 | 11 | 31% |
| Murrupula | 13 | 41 | 10 | 77% |
| Total | 68 | 157 | 40 | 59% |

**Table 6. CDFs pre- and post- training test results.**

| District | Average Pre-test score[a] | Average Post-test score[a] | Average change between pre and post-test |
|---|---|---|---|
| Erati | 7.44 | 9.46 | 3.53 |
| Mecuburi | 4.46 | 9.13 | 4.61 |
| Mogovolas | 5.03 | 8.21 | 3.18 |
| Murrupula | 5.62 | 7.65 | 2.33 |
| **All districts combined** | **5.64** | **8.61** | **3.41** |

[a] Maximum score: 12

'communities', and the absence of a consensual and reliable list of communities at local level. We have therefore not been able to determine the true geographical reach of the intervention.

**1.3 Dose delivered.** *1.3.1 CDFs trainings.* CDFs were trained at district level in a series of three-day training sessions of about 25 participants each, by a consultant trainer and the respective district NTDs Focal Point, who had previously attended a five-day training of trainers. The trainings were all completed by the end of August 2014.

FGDs conducted with CDFs showed a high satisfaction with the training content among participants. However, in midterm FGDs, they expressed initial concerns about not receiving compensation for their role as a CDF and felt that the training was too short for them to feel fully comfortable conducting dialogues on their own.

Overall, evaluation of the trainings showed a sharp increase in knowledge gained by participants regarding schistosomiasis prevention and treatment measures when comparing pre- and post-training test results presented in Table 6.

Monitoring data indicated that most of those trained appeared still active at the end of the study (until February 2016, three months after the second cycle had ended). All facilitators interviewed at endline confirmed they had been actively conducting CDs throughout both cycles. A few of them in two districts reported knowing about facilitators who had stopped.

*1.3.2 CDs conducted.* It was not possible to establish the total number of CDs conducted across the intervention areas due to some CDFs either not returning or not filling in the feedback forms and planning sheets. Based on a total of 1,458 CD feedback forms collected from 90% of the communities (as per the list based on census enumeration areas) with at least one CDF trained, on average about 30 CDs were conducted during the intervention period per community, which is broadly in line with the number expected, assuming an average of two to three CDFs per community and twelve CDs for the project period.

*1.3.3 CD attendance.* The analysis of feedback forms collected and the qualitative FGDs conducted at midterm and endline consistently indicated that, on average, the number of participants per CD comprised between 25 to 45 participants, and was very variable, ranging from under 20 to over 100 participants in some instances. Facilitators often attributed good attendance to community and religious leaders spreading the word and inviting people to attend.

> In my community, the church leaders gave us a lot of support, many people from the church came to participate in the community dialogues. They liked it and they like it because they participated actively and followed what we explained. The people of the church were very supportive.

> CDF, FGD, endline

Often CDFs would see a core group of regular attenders and many who attended only occasionally were attracted by word of mouth of participants in previous dialogues. Both CDFs and CD participants accounts indicate that more women than men tended to come to the dialogues. This was mainly attributed to women's presence in the community, contrary to men who were away from home during the day because they were engaged in income generating activities, such as fishing and farming.

**1.4 Local adaptations of the intervention.** *1.4.1 CDFs' catchment area.* According to most CDFs interviewed and in light of the ambiguous definition of the term community, the majority of them understood that they were responsible for an area larger than their usual community of residence. They usually tried to cover their own community of residence as well as reaching out to nearby communities depending on their perception of their respective catchment area, by organising CDs in various locations (e.g. mosque, church, market), inviting participants from neighbouring communities to participate in CDs, and even organising CDs outside of their usual area of residence in an effort to maximise reach. Indeed, from the feedback forms received, we noted that some communities where no CDFs were trained actually received dialogues on an ad hoc basis, mainly in Mogovolas district. CDFs however reported that they were unable to deliver CDs in all communities within their vicinity due to long distances and access challenges, including the lack of transportation.

> In my district, we primarily focused on the communities around the health centres, this is where the Dialogues happened. . . Some communities are remote and have not been participating in Community Dialogues, these are still not aware of what is being discussed or have not yet information. I think maybe the radius was a little small, because the (name) district is a huge district. We would like the Community Dialogue to reach out to the peripheries or to the most remote areas.

> KII, endline

The gap in coverage was considered a weakness of the approach and both CDFs and CD participants interviewed were of the opinion that it was crucial that the dialogues reach all communities to avoid any perceptions of discrimination or exclusion.

In FGDs, CDFs frequently mentioned that they cooperated and facilitated the dialogues in pairs or groups of CDFs. This cooperation was used as a coping strategy to remediate to this perceived weakness, despite pointing at organisational and logistical challenges, such as distance between their respective residences that hampered easy communication and meetings.

*1.4.2 Topics discussed.* From the feedback forms, CDFs appeared to cover the main themes from the suggested list of topics; however, it appears that topics were often discussed together in 'overview' sessions instead of separate CDs. Observation visits also noted that CDFs did not manage to focus the discussions on specific topics within the flipchart with a tendency to rather present and discuss all themes of the flipchart in each dialogue.

The decisions made during CDs generally corroborate the practices recommended in the flipchart. Discussions were also initiated around other health issues, beyond schistosomiasis, which were felt by participants as common, relevant, but not sufficiently addressed by health interventions.

Most frequently, a diverse range of hygiene measures were discussed, such as: maintaining the cleanliness of households; washing of dishes; food hygiene and storing and personal hygiene. Other diseases commonly addressed included malaria, cholera, diarrhoeal and skin diseases (ex: ringworm, tungiasis, bed bugs). Few respondents also recalled having discussed sexually transmitted diseases during the dialogues.

We discussed a lot of things, not just what is in those books but also those issues related to compound hygiene, diarrhoea, as well as washing dishes before and after eating.

CDF, FGD, midterm

*1.4.3 Support to CDFs*. Linking CDFs with their district-level supervisor did not function as planned. While the intervention provided phone credit to CDFs to contact their district-level supervisor, few of them actually used this to reach out to the district Focal Points. According to CDF accounts collected at midterm and endline, they felt uncomfortable "*jumping over hierarchies*" by contacting the district Focal Point directly instead of liaising with the nearest health centre; some also mentioned practical considerations such as not having a personal mobile phone.

They also reported that district-level Focal Point was often not available at the rare instances when they tried to reach out.

I tried to call Dr. [name] once, but he said that "now I'm on the street, I can't answer, call me another time". That done, I called another time and the phone was out of range.

CDF, FGD, endline

Instead, CDFs indicated that they would have preferred supervisors to visit them on site, which was considered important to reinforce their credibility to the wider community and a sign of appreciation for their volunteer work.

**1.5 Mechanisms of impact.** *1.5.1 Ten-step methodology*. Most interviewed and observed CDFs had good recall of the ten-step methodology to plan and lead CDs, which they perceived as necessary for a successful dialogue and helpful in order to avoid using lecture style. Observation visits reports, however, seem to indicate that the ten steps were not always strictly followed. Despite this, both CDFs accounts and observation visit reports confirm that the critical steps before (linking with community governance structure) and during (explore the topic and identify actions) the CDs were effectively conducted.

CDFs appropriately linked up with their respective community structures upon returning from the training and in the planning and organisation of all CDs. CDFs were of the opinion that community leaders played a crucial role in both preparing and conducting the CDs: it was mainly the community leaders who mobilised community residents to participate in the CDs and supported the identification of communities' needs and problems, the decision-making process and the actual monitoring of decisions reached.

With the help of a community representative, like a leader, a queen, a leader or a secretary, we would talk and set the date, time and place. Then the representatives disseminated the information to the communities, because a facilitator alone without the representative will not be able to mobilise everyone in the communities.

CDF, FGD, midterm

During the sessions we were supported by the community leaders, who walk in the communities to ask if people have latrines and bathrooms. . . They also asked the communities why they do not have the latrines. Then, a follow-up was made for those who do not have latrines to build.

CDF, FGD, endline

*1.5.2 Exploring the health topic.* Across all four districts, the exploration phase of the CD appears to have been appropriately conducted. Discussions were described to have developed around questions posed by the facilitators to explore the themes, with participants sharing their knowledge, experiences and doubts. Discussions often revolved around the disease symptoms, similarities and differences with other diseases, and the treatment and prevention options, as per the intended topics. Observation visits' reports confirm that facilitators corrected misconceptions and discussed with participants until a consensus was reached.

> The facilitator explored the topic, for example: "Today, as always we will talk about bilharzia. First of all, I'd like someone to remind me of what bilharzia is. . . do you know what it is?" Members of the community responded correctly. One man said that "it is a disease that is caught in the still waters where there are snails with bugs". A woman raised her voice and said that "One can get Bilharzia when walking in dirty places, where people defecate and urinate without any control and sanitary care." The ideas shared by the first two persons were reinforced by others, showing that everyone agreed with the idea.

> Annotation from observation visit of CD session, district A, second cycle

A large majority of community respondents who attended CDs, both in midterm and endline FGDs, showed good levels of correct biomedical knowledge on how the disease is acquired, transmitted and prevented. They were able to identify the disease signs and symptoms, and described quite accurately the roles of poor hygiene, contaminated freshwater, snails and microorganisms in the disease transmission cycle.

> Bilharzia is a disease caused by lack of hygiene, doing necessities in the river, urinating in the river, going fishing without boots, washing without gloves or without shoes, then snails that are there on the river bank and they also have some animals that we cannot see, and those when it enters your body makes you suffer; that illness which we had not heard about, (we) used to say that it is a spell or something that came out of your body, we would take the person to the hospital but did not understand anything, and now that we heard that, we arrive at the hospital and we really tell the truth, how it happened and recognise this disease.

> CD male participant, FGD, midterm

Respondents could correctly identify behaviours that put people at risk of infection; and demonstrated good understanding of recommended preventive and control practices as presented in the flipchart.

CD participants identified the CDs as their primary source of information about the disease and highlighted how dialogues had allowed for correction of misconceptions.

> They thought that this disease is not acquired by staying in dirty water. They said bilharzia was a disease that was transmitted itself through heredity. So, all children were to get this illness because the ancestors also had it. They thought it was transmitted through incest, when someone had sex with a relative. Then we explained how it is acquired, which is through the dirty waters in the river, the lack of hygiene and soon they come to believe.

> CDF, FGD, midterm

Some participants however still held misconceptions about how the disease is transmitted. As an example, one participant indicated that the snail enters the body rather than the parasite.

Some CD participants thought that schistosomiasis could be acquired by standing in someone's urine when in a latrine or by stepping in someone's urine in the bush.

*1.5.3 Identifying actions to resolve issues.* The following phase of identifying actions was often not distinctly conducted, but rather merged with the exploring and decision-making phases, but this did not impede the identification of issues. In endline FGDs, CD participants reported discussing social norms and identifying practices that were not conducive to their health. Likewise, modelling of behaviours was performed with many of the respondents describing reaching a consensus on appropriate practices to be adopted.

> Mostly, we talked about decision making, but we were not the ones who made the decisions. But from the discussion, the community itself made decisions for itself. So we discussed how to be aware of bilharzia. If someone is sick you should take medicines, go to the hospital. Therefore, we discussed and reached a consensus that from now on we will, for example, dig latrines, build bathhouses. After that, we made action plans.

> CDF, FGD, endline

*1.5.4 Decision-making on individual and communal actions to be implemented.* In most CDs, the decision-making phase was the one that received the least emphasis. The decisions recorded in the CDFs monitoring tools mostly reflect effective prevention and control behaviours outlined in the visual materials and cover mainly generic hygiene practices, construction of latrines and a commitment to participate in MDA campaigns. Most of these actions are individual though some communal actions were also identified such as the construction of wells and boreholes, and in few instances also building latrines and bathrooms in the communities. However, the course of action was often left vague.

While all CDs resulted in clear identification of actions, the detailed planning of who was going to do what, when and how to ensure that these actions were implemented appeared to be more challenging. Often, the implementation was left to each individual's responsibility or to the community leader to enforce. In a few instances, there are indications that community-led mechanisms for enforcing decisions existed and were applied, such as: community leaders and CDFs conducting house-to-house visits to check on the implementation of commitments made at the dialogue or the community organising itself to build latrines, with specific tasks attributed to groups of people.

> In my community a committee was set up to monitor latrines (construction). All houses must have latrines, they must clean the compound, and when the latrine is full we must control the level well so that we can bury it . . .

> CD male participants, FGD, midterm

> The secretary (of the community leadership structure) would mobilise the people and send the community police to go around the rivers, and check who is bathing in the river to take them out of there.

> CD participants, FGD, endline

Existing community structures such as the administrative and traditional leaders, community health committees, community police, community health workers, and activists were involved in implementation and monitoring mechanisms, where they existed. Youth and religious leaders are also mentioned in several instances with a role of information dissemination

both to communicate the date of the next CD and also to raise awareness of community members on the decisions made and the necessity to collectively comply with these.

While most decisions reached through CDs were fairly generic and relied on individuals to implement (i.e. everyone should build a latrine), due to the community leaders' involvement in the CD and endorsement of the decisions, these commitments were often turned into a community rule applicable to all community residents, paving the way for protective behaviours to be shaped into social norms, as community-recommended practices.

*1.5.5 Participatory discussion*. Most dialogues followed the inclusive participatory discussion method as intended. While supervisors considered that some observed CDs took the form of question and answer sessions, respondents in FGDs were of the opinion that CDs were largely participatory discussions with opportunity for participants to share their views and experiences and put forward questions.

> They also said that everyone has the opportunity to speak, no one can feel excluded and they said that this is not a political party because these are diseases that affect us. . . all people have to speak about what they feel in their body, whether children or adults, everyone at the meeting had the chance to speak.

> CD male participants, FGD, midterm

District-level health staff interviewed were of the opinion that the participatory nature of discussions made the dialogues more effective at changing knowledge and attitudes of people than lectures given by health workers.

## 2. Acceptability of the community dialogue in the prevention and control of schistosomiasis

The qualitative data indicated that the intervention was highly acceptable to those involved. The CDA was considered by respondents as highly relevant, engaging and enlightening.

**2.1 Participants' satisfaction.** *2.1.1 An engaging platform*. Interviewed facilitators and district health staff were of the opinion that the CDA facilitated a better understanding of health issues as information was transmitted "*from the community to the community*", compared to health talks given by health staff. Facilitators valued the CDs as they allowed community members to share knowledge, experiences and ideas, and together they discussed and reached a consensus on how to overcome schistosomiasis.

> Then each one starts to talk about what affects most the community because the dialogue is meant to reach a consensus. Since in the Dialogue we learnt many things that we did not know before, because we were accustomed to lectures. In the lectures, we are only told that disease x can be prevented in this and that way. While in Community Dialogues each one gives his contribution to a particular fact.

> CDF, FGD, endline

Some CDFs reported an initial mistrust of community members who associated the purpose of CDs with political parties' concomitant election campaign and thus did not participate in the CDs. However, this seems to have been clarified during the first CDs and disseminated with support from community leaders, resulting in more interest and more participants. The positive feedback to the community about the dialogues also led to attracting new participants through word of mouth from the CD participants. This confirms the acceptability of the approach.

Also those who participated communicated to others what they learnt and reported that those who did not participate missed. So this motivated many participants to join Community Dialogues.

CDF. FGD, midterm

*2.1.2 Satisfaction driven by knowledge gains.* The CDs were considered by all respondents as a good platform for the transmission of health information. Participants appreciated the gains in knowledge the CD provided, and enjoyed subsequently teaching their children, other family members and neighbours. Likewise, CDFs enjoyed the CD as it enabled them to learn and teach about behaviours that are conducive and detrimental to health.

We like to learn. Because I have children, I like to learn and then teach my children.

CD male participants, FGD, midterm

Respondents felt that dialogues were highly relevant because they helped them to fill gaps in their knowledge concerning a disease which affects them. Participants commonly described that before the CDs, "*we were in the dark*" (Note from observation visit, first cycle). All the information addressed during CDs was appreciated, but people especially enjoyed learning about the transmission and prevention of schistosomiasis, as well as learning that schistosomiasis has a cure and can be treated at local health facilities. People highlighted that they liked the opportunity to correct their misconceptions, and that by learning about hygiene measures they also learnt to prevent other diseases.

I liked it because when we go to Community Dialogues we receive information on wellbeing and communicate to other people in the community who could not be there. Because they (Community Dialogues) are reducing many diseases in the community. In the old days we were lost, we knew nothing.

CD participants, FGD, endline

We liked it because we did not know how the disease is acquired. In the old days we thought that this disease was contracted through having sex with a woman; others thought it was hernia. But with the explanation we learnt that just noticing the signs x and y sign means it is bilharzia, and it is acquired by staying long in the water, where we sit, where we fish.

CD participants, FGD, endline

*2.1.3 Appreciation of the visual flipchart.* CDFs described the flipchart and guidebook as essential tools for conducting participatory sessions, which assisted them with the recall of information learnt during training and were used to clarify questions and doubts arising in the course of the discussions.

The guidebooks are our hoes for the farm. Without the guides we could not work.

CDF, FGD, endline

The visual flipchart in particular was much appreciated both by participants and facilitators of CD sessions. The illustrations provided validation of the information transmitted by the facilitators. Participants said that if it wasn't for the flipchart, they wouldn't have necessarily believed in the information shared by the facilitators.

Also in my community it is like this, people do not believe without showing them the images. That's when they opened the album and showed the pictures, only then they believed.

CD male participant, FGD, midterm

The flipchart's illustrations were positively appreciated because they reinforced facilitators' messages and reflected local context and practices. It was frequently described how during the dialogues, participants identified themselves or people they knew with the scenes pictured in the flipchart.

So that was the job that the flipchart did, it helped a lot because people saw with their eyes and when we talked we all screamed. . . that's [name] of our community!

CDF, FGD, midterm

**2.2 Relevance and applicability of protective behaviours.**   Most protective behaviours discussed during CDs were perceived as relevant and applicable by respondents. Respondents' accounts show that gaining knowledge of risk behaviours and protective practices also translated into changes in attitudes, and actions to treat and prevent the disease. Respondents shared the perception that the CDs had resulted in the adoption of preventive practices, reduction in the burden of schistosomiasis, and improvement of people's health in general. They all were of the opinion that CDs should continue so as to sustain these changes.

In my community they thanked us and we want this project to last for a long time since it is helping the community. Since last year we noticed a change in the community, people are already healthy. We ask for a lot of help at the same time, continue to hold meetings to explain to the community because if we stop we will go back to what we used to do before.

CD male participant, FGD, midterm

*2.2.1 Shift in care-seeking practices.* During the FGDs, CD participants described that people had learnt about the importance of seeking immediate care at health facilities upon presentation of signs and symptoms of schistosomiasis. The majority of the respondents stated that most people's attitudes had changed with health facilities now being the first choice in care-seeking, in place of traditional doctors. CDs seem to have increased awareness and understanding of the diagnosis and treatment available at health facilities and thus their self-efficacy in seeking medical care.

In the old days we used to think we should turn to the healer. When we got sick, of course, with a big belly we thought we were charmed—soon, we would turn to the healer. We used to say this is not bilharzia, he was infected by a woman. Hence the healer would give him a number of roots, he would defecate various things and never improve. After we had the Community Dialogues, it is now difficult to find people at the healer. They learnt that when they are ill they should go to the hospital to take pills. We did not know about bilharzia, when we saw someone with pale skin and pale hair, we thought he was impregnated, things of African magic.

CDF, FGD, endline

Interviewed district-level health staff confirmed that dialogues helped people to better artic-ulate their symptoms which in turn facilitated the interactions with health workers in the diag-nosis and treatment process. However, community respondents often pointed at inconclusive treatment experience citing the lack of medicines and diagnostics at public health facilities.

Community-level respondents expressed improved awareness about MDA process and purpose, but also frustration, in some communities, about the constant delay or cancellation of announced MDAs.

> Some people held the perception that treatment campaigns killed people. During the dis-cussion, the facilitator asked the participants in the Dialogue who had already lost a relative or acquaintance during the campaigns. It was proved that the idea that campaigns killed people was an invention of some people in bad faith.
>
> Note from observation visit, second cycle

*2.2.2 Hygiene and water-handling practices.* In terms of prevention, the most cited changes included building latrines and bathrooms, cessation of bathing in freshwater, washing laundry and dishes away from water ponds, treating water, and generic 'hygienic' practices such as hand washing.

While people had previous knowledge about the importance of treating water before drink-ing, respondents reported that discussing this issue during CDs had changed peoples' attitudes from considering boiling water as too laborious towards recognising its value.

In terms of avoiding contact with infected water, most common reported preventive prac-tices learnt and adopted were the prohibition of swimming and playing in water ponds, and the collection of water for washing and bathing away from the freshwater sources.

> In the old days, we the mothers before we knew we used to take our children to shower, arriving at the river we would let the children to swim in the river, while we were washing the clothes; But, in this water there are microbes that enter our organism and we get bilhar-zia; so we ended up seeing children pee weird things; Now we know that we should prevent this by building bathroom, when we go to the river we must take buckets to get water and wash the clothes outside, when we go we should draw water and bathe in the bathroom, so let's prevent this disease.
>
> CD participant, FGD, endline

*2.2.3 Practices with limited applicability.* Construction of latrines was the change most emphasised by respondents, however the correct use of latrines was seldom addressed. Respondents reported that only a few community members had the resources to construct improved latrines resulting in poor latrines (for example without proper slab or ventilation) limiting their actual use.

Despite the use of boots being one of the most frequently raised topics during discussions and participants being aware of their importance in avoiding contact with infected water when fishing or farming, the use of protective gear was only reported by few respondents. Most cited lack of financial resources to acquire boots.

> In my community, they asked me about the boots issue, they said: "since we have this dis-ease in our fields and we don't have boots, shall we stop farming? How are we going to live? (. . .) In relation to this concern, I felt limited.
>
> CDF,FGD, endline

*2.2.4 Suggestions for improvement.* Both facilitators and participants noted that the weak point of the CDA was the challenge in reaching out to all communities. They recurrently referred to the fact that other neighbouring communities needed to have access to and benefit from the CD platform.

Facilitators felt they were made responsible for too many communities without being provided with the means to actually reach these, both in terms of support in transportation and also in terms of compensating for their time. They were of the opinion that more facilitators needed to be trained in order for each facilitator to be responsible for their own community and all communities being reached.

Suggestions for improving the approach revolved around additional support for the CDFs especially: identification material, such as uniforms or badges, monetary incentives to compensate for their work, and transportation so that CDFs can reach more villages.

> We would like to have a badge like other projects do. When you have a badge there is great respect in the society. When the person passes by, people already know, you don't even need a diploma when you have a badge. Because it does not lie anyway.
>
> CDF, FGD, endline

The lack of monetary incentive was the most frequent suggestion from CDFs, as this was considered a motivating factor and a form of recognition by the health system. More frequent supervision visits and refresher trainings were also cited for improvements. CDFs and CD participants were of the opinion that CDFs should be involved in community-level distribution of treatment for schistosomiasis, either through active participation in MDA delivery or provision of treatment to compensate for lack of access to health facilities.

## Discussion

The intervention was found to be feasible, generally delivered as planned, and highly acceptable to all respondent categories.

### Feasibility of using the community dialogue in the prevention and control of schistosomiasis

Non-specialist community volunteers, who received only a short training course and minimal follow up, were able to autonomously and regularly deliver CDs, working consistently with existing community leadership structures.

Despite not receiving financial incentives, the CDFs remained active throughout the project. As found in other studies [17, 35], social status in the form of recognition (i.e. as being useful to their community and self-fulfilment (i.e. gain new knowledge useful for themselves/family) and social influence act as important non-monetary incentives.

The intervention could be further improved by better defining the geographical coverage expected from the CDFs, referring to locally recognised and understood concepts of community. In planning for CDF selection, systematic listing of communities to be reached should be conducted in close collaboration with local stakeholders to agree on a definition of the 'community', the unit for the intervention. In defining the area to be covered by CDFs, distances between villages or within a community cluster should be considered as reaching to neighbouring communities is likely to involve transport and opportunity costs that CDFs are often unlikely to be able to afford in resource-poor settings.

Although the recommended three-phase process of exploring, identifying issues and making decisions was often not strictly followed during dialogues, CDFs demonstrated the capacity

to facilitate participatory discussions that sought local experience, and enabled analysis, sharing and decision-making, which are essential elements of participatory learning and action approaches [27].

CDFs' facilitation skills could be further strengthened to strengthen the implementation of the ten-step methodology through regular practice coupled with supportive supervision, or through longer training and support supervision, as demonstrated by other projects [28], but the latter approach may involve incremental implementation costs.

Participants praised the participatory nature of the discussions, and particularly appreciated the fact that the exploration of issues, and identification and implementation of actions were locally owned and led.

CD participants described how the dialogues, through the exploration phase which builds on local understanding, helped them to correct misconceptions, particularly in relation to heredity and sexual transmission, which are common misconceptions found elsewhere in sub-Saharan Africa [36–38].

## Acceptability of the community dialogue in the prevention and control of schistosomiasis

All respondents considered the CD as a good platform for the transmission of health information, which was perceived as highly relevant because dialogues facilitated knowledge gains concerning a disease which affects them.

CD participants demonstrated a fairly accurate and detailed understanding of the disease causes, symptoms, risk behaviours and prevention and treatment mechanisms, and explained how the dialogues allowed them to learn new insights. These gains in knowledge were confirmed by a post-intervention quantitative KAP assessment at population level which found improved knowledge of risk behaviours and disease's signs, symptoms, and treatment across the four districts where the dialogues were implemented compared to baseline [22].

CDFs as members of the community, know the languages, customs and context, and are able, when trained, to deliver health messages in a culturally appropriate manner easily understood by local people [39]. Both participants and CDFs pointed at the flipchart's visuals as key aides for learning. The fact that the flipchart and guidebook focused on delivering only key biomedical information and translated a complex disease transmission cycle into a series of risky versus protective practices was key: it provided relevant information in a format suitable to the low health literacy levels of the target population.

Most protective behaviours discussed during CDs were perceived as relevant and applicable by respondents. The CDs triggered relevant individual and communal actions towards improved disease prevention and treatment in participating communities. Actions decided upon during CDs often then applied to every community resident. This process positioned correct prevention and control practices as the community recommendations, indicating that the CDA can shape social norms around prevention and treatment mechanisms, which is a determinant factor for changing behaviours [40, 41].

Respondents shared the perception that the CDs had brought some positive attitude and behavioural changes in their communities in relation to schistosomiasis understanding, prevention and management, particularly in terms of care-seeking, willingness to comply with MDA, sanitation practices, and avoiding contact with infested water. This was confirmed by the endline quantitative KAP assessment at population level which found improved positive attitudes towards preventing the disease compared to baseline, with those reporting actively doing something significantly more likely to cite an effective behaviour at endline [22].

While some CD participants may not have a full understanding of the details of the disease transmission cycle (i.e. the snail entering the skin instead of the parasite) this does not seem to undermine their motivation and capacity to apply the recommended prevention and treatment practices, particularly in terms of hygiene and sanitation behaviours. Sacolo et al. note that individuals with higher knowledge are more likely to adopt protective behaviour towards schistosomiasis infection, and recommend that comprehensive schistosomiasis-related knowledge, including information pertaining to the life cycle and types of schistosomiasis, should be standardised and integrated as a key component of national schistosomiasis control programmes [36]. Our experience shows however that providing community members who have a low-level of education, with limited but locally relevant information, which responds to their essential information needs, may be sufficient in a first phase to build their capacity to engage in recommended prevention and control practices. Also, the intervention materials built on existing messages and visuals used in the country at community level to promote basic hygiene and sanitation recommended for other diseases' control, such as soil transmitted helminths and diarrhoeal diseases. Framing and aligning recommended behaviours with a cluster of health-protective practices, to which people have been previously exposed through other health promotion programmes, may also be an important enabling factor.

## Embedding CD into disease control programmes

The CDA provided a platform for community leadership and community members to put the topic of schistosomiasis on the community agenda and to discuss it at length until reaching consensus on individual and communal actions.

The CDA demonstrated potential for improving communities' engagement in wider schistosomiasis prevention and control through the use of participatory techniques. On the continuum of community participation proposed by Draper et al. [11] ranging from mobilisation to collaboration and empowerment, the CDA can be located at the lower level of participation, described as community mobilisation: it involves enabling selected community members to conduct participatory CDs, while topics and tools were predetermined outside of the community.

However the feasibility, acceptability and sustainability of the CDA may be compromised in the long-term if the quality and consistency of service delivery (both for MDAs and case management) does not meet communities' expectations, and if no complementary strategies are established to provide contextualised alternative solutions to the recurrent lack of access to safe water and basic sanitation.

The intervention prompted care-seeking and raised demand for treatment among CD participants, who reported they actively sought care at health facilities after attending dialogues and were readily waiting for the MDA. This confirm results from a previous process evaluation of the CDA applied to community-case management of childhood diseases in terms of impact on shifting care-seeking intentions towards qualified health providers [20]. Other studies also noted that knowledge of the MDA and of the disease transmission pattern were prominent factors associated with motivating compliance and increasing coverage for MDA for lymphatic filariasis [42]. However, respondents expressed disappointment that diagnostic and treatment services for schistosomiasis were not always readily available at health facility level when they sought care. Although CD cycles were meant to coincide with MDA campaigns, these were not implemented as planned, and there was no formal linkage between the intervention and MDA coordination and delivery to adjust the content and timing of the CDs better to the actual service delivery. This was perceived as a weakness both by participants and CDFs as it created a demand for a service which could not be fully met.

Respondents repeatedly referred to the lack of resources (i.e. boots, safe water point) as the main barrier to effective behaviour change, particularly in terms of improved sanitation and avoiding contact with infected water. Indeed, the intervention did not include the provision of alternative safe water sites and sanitation facilities, which remains a basic need of communities living in infested areas, as noted by several authors [36, 43, 44]. King underlined the extremely important role of health education to trigger significant societal change in patterns of water use and sanitation [4], but also warned that health education alone may have only a minimal impact on transmission, due to remaining limitations in choices for safe water use. As put by Bardosh, 'the same poverty-inducing factors that drive NTDs transmission present various context-specific challenges to controlling them' [19].

The CDA would thus benefit from extended linkages with other community development schemes in order to mobilise resources to overcome barriers to effective implementation of decisions and improved practices.

An important element of the CDA that needs to be strengthened is the degree of embeddedness of CDFs within the health system. Establishing formal linkages between CDFs and the nearest health facility, instead of the district level, would provide opportunities for supporting CDFs and linking them with health service planning and delivery. Enrolling the CDFs in local planning around MDA and in conducting targeted social mobilisation activities would likely strengthen uptake of MDA. This would also strengthen their positioning as the NTDs focal person at community level. CDFs could provide useful feedback and community perspectives on MDA and NTDs control issues to programme managers, which could be further integrated into future programming [39]. A recent paper describing the development of a CDA intervention for addressing the drivers of antibiotic resistance in Bangladesh paid particular attention to embedding the intervention within the existing health system, arguing that this provides a valuable and potentially sustainable entry point for subsequent scale-up [45].

The fact that the CDA can build on existing social infrastructure, and does not require strong external agency back-up beyond the initial training and materials, provides opportunities for integration into routine tasks of local health services, replication and scale-up to other endemic regions in the country, speaks in its favour. A programme in Cameroon showed that when culturally appropriate health education is delivered at the community level together with capacity to diagnose and treat schistosomiasis, then the positive changes in knowledge also translated into behaviours, with encouraging results on infection control [46].

Following regular updates on study progress and a presentation at a national stakeholder meeting, the National NTDs Programme created a social mobilisation committee to strengthen and guide community engagement efforts within the existing NTDs strategy, and developed plans to train all provincial NTDs Focal Points in Mozambique on the use of the CDA. To assess the sustainability of scale-up plans, a thorough political economy analysis would be needed to identify the key elements required for embedding such a CDA into the NTDs control programme and within the local health system landscape, taking into account resource-limitations, working norms, and local roles and responsibilities in community health.

## Strengths and limitations

This study used primarily qualitative and process data and bears certain limitations. The low literacy level of CDFs has meant that routine monitoring did not result in rich data which limits the depth of analysis of CD feedback forms and CD planning sheets. Primary qualitative data was collected in the local language and then translated into Portuguese by the Research Officer. Only quotes originally in Portuguese in the data set have been translated into English

for this publication; these quotes were not back translated into local language to check for optimal accuracy. This process may have led to misinterpretation, loss of information and bias due to translators' interpretation and assumption, as is often experienced in qualitative research [47].

Using convenience sampling in selecting the study sites for midterm and endline FGDs and for observation visits is another limitation because some research participants from remote areas might be missed. We have addressed some of these gaps through triangulation of information from different groups, which is a useful strategy for checking consistency within and across the groups [48].

The data presented is mainly self-reported hence likely to reflect some desirability bias from respondents that may limit the accuracy of measurements in knowledge and behaviours. In focus group discussions, in some instances outspoken individuals can dominate the discussion, thus results may not be generalisable to the larger population [49].

The evaluation was conducted after only one year of intervention's introduction, which might be too short time to effect people's behaviours.

Finally, while the intervention aimed to address a number of cognitive, motivational and social determinants, the actual uptake among communities of the behaviours targeted by the intervention also depends on a range of other individual and contextual factors [12, 50] which may include: specific psychological or skills-based barriers, the availability and quality of diagnosis and treatment services for the target disease at facility level, the delivery of MDA, and the availability of the basic resources for people to practice preventive behaviours, such as soap for hand washing, safe water, and skills on how to wash hands, build and maintain latrines, and treat and handle water safely.

## Conclusions

The CD intervention was found to be feasible and highly acceptable. Non-specialist community volunteers, who received only a short training course and minimal follow up, were able to autonomously and regularly facilitate participatory CDs. Participants particularly appreciated the fact that the exploration of issues, and identification and implementation of actions were locally owned and led. The approach and tools used in CDs were suitable to the low health literacy levels of the target population, facilitated knowledge gains among participants and triggered relevant individual and communal actions towards improved disease prevention and control in participating communities.

Within a relatively short timescale of one year the CDA has shown potential to increase knowledge, shape social norms and prompt communities to take locally relevant actions to contribute to disease prevention and control. Such an approach should be embedded within disease control programmes and the health system to create long-lasting synergies between the community and health system for increased effectiveness. However, for behavioural change to be feasible, community engagement strategies need to be supported by improved access to treatment services, safer water sources and basic sanitation.

## Supporting information

**S1 Appendix. Community dialogue training manual.**
(PDF)

**S2 Appendix. Schistosomiasis flipchart.**
(PDF)

**S3 Appendix. Community dialogue guidebook.**
(PDF)

**S1 Table. Coding frame.**
(XLSX)

# Acknowledgments

We gratefully acknowledge the support received from the Republic of Mozambique's Ministry of Health and the Provincial Health Directorate in Nampula province. In particular, we would like to thank Dr Francisco Mbofana, former Director of the National Public Health Department for his feedback. We would also like to thank our colleagues at Malaria Consortium for their ongoing guidance and support, in particular Ana-Cristina Castel-Branco, and Diana Thomas, who critically proof-read an advanced draft of the paper. Finally, we would like to acknowledge the kind support of the study participants.

# Author Contributions

**Conceptualization:** Sandrine Martin, Christian Rassi, Kirstie Graham.

**Formal analysis:** Sandrine Martin, Christian Rassi, Valdimar Antonio, Jordana Leitão.

**Funding acquisition:** Christian Rassi, Rebecca King.

**Methodology:** Christian Rassi.

**Project administration:** Christian Rassi, Valdimar Antonio, Kirstie Graham.

**Supervision:** Sandrine Martin, Christian Rassi, Valdimar Antonio, Ercilio Jive.

**Validation:** Sandrine Martin, Christian Rassi.

**Writing – original draft:** Sandrine Martin, Christian Rassi, Jordana Leitão.

**Writing – review & editing:** Christian Rassi, Valdimar Antonio, Kirstie Graham, Jordana Leitão, Rebecca King, Ercilio Jive.

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
