## [Decision Letter · Decision Letter 0]

13 Mar 2020

PONE-D-19-34788

Engaging affected communities in the prevention and control of schistosomiasis: evaluating the feasibility and acceptability of a Community Dialogue intervention in Nampula province, Mozambique

PLOS ONE

Dear Ms Martin,

Thank you for submitting your manuscript to PLOS ONE. After careful consideration, we feel that it has merit but does not fully meet PLOS ONE’s publication criteria as it currently stands. Therefore, we invite you to submit a revised version of the manuscript that addresses the points raised during the review process.

I would particularly urge you to consider the request for clarity and focus expressed by Reviewer 2, which I fully support. I accept that you have previously published on some of the key debates and topics related to community dialogue, but equally I feel that each paper should have enough information to be useful to the reader on its own merits. I also endorse the reflections by the reviewer regarding some of the language included in the paper (ie, 'fragile state' etc).

We would appreciate receiving your revised manuscript by Apr 27 2020 11:59PM. To enhance the reproducibility of your results, we recommend that if applicable you deposit your laboratory protocols in protocols.io, where a protocol can be assigned its own identifier (DOI) such that it can be cited independently in the future. For instructions see: http://journals.plos.org/plosone/s/submission-guidelines#loc-laboratory-protocols

We look forward to receiving your revised manuscript.

Kind regards,

Enrique Castro-Sánchez

Academic Editor

PLOS ONE

Journal Requirements:

2. Please include the registration number for the clinical trial referenced in the manuscript.

4.  We note that [Figure 2] in your submission contain map images which may be copyrighted. All PLOS content is published under the Creative Commons Attribution License (CC BY 4.0), which means that the manuscript, images, and Supporting Information files will be freely available online, and any third party is permitted to access, download, copy, distribute, and use these materials in any way, even commercially, with proper attribution. For these reasons, we cannot publish previously copyrighted maps or satellite images created using proprietary data, such as Google software (Google Maps, Street View, and Earth). For more information, see our copyright guidelines: http://journals.plos.org/plosone/s/licenses-and-copyright.

1.     You may seek permission from the original copyright holder of Figure [2] to publish the content specifically under the CC BY 4.0 license.  

Reviewers' comments:

Reviewer's Responses to Questions

**Comments to the Author**

1. Is the manuscript technically sound, and do the data support the conclusions?

Reviewer #1: Yes

Reviewer #2: Partly

2. Has the statistical analysis been performed appropriately and rigorously? 

Reviewer #1: N/A

Reviewer #2: N/A

3. Have the authors made all data underlying the findings in their manuscript fully available?

Reviewer #1: Yes

Reviewer #2: No

4. Is the manuscript presented in an intelligible fashion and written in standard English?

Reviewer #1: Yes

Reviewer #2: Yes

5. Review Comments to the Author

Reviewer #1: This is a well-written description of a community dialogue approach used to improve community-wide knowledge, attitudes and engagement in schistosomiasis control activities. The current manuscript leverages both qualitative and quantitative data collected during process evaluation with reference to pre-versus post household survey comparisons.

The manuscript describes well key findings regarding the implementation outcomes of feasibility and acceptability especially regarding fidelity to the intervention, dose, reach and impact. As is typical with community participatory interventions, it is difficult to report any improvements in hard outcomes e.g. reduction in prevalence of S. haematobium. This might need a randomised control trial approach and a longer follow up period.

The jury is still out there regarding the utility of community participatory approaches like CD in improving specific health outcomes. For this reason, more hybrid effectiveness-implementation research studies are required as opposed to simple effectiveness studies. Given this description of the implementation approach and suggestions towards improved community knowledge, attitudes and practices, I hope that the authors consider a rigorous follow up study (maybe type 2 or type 3 hybrids?) to bulk up the knowledge base on the utility of community participatory approaches.

I have only minor editorial concerns that could be addressed by the authors. Please check the consistent use of acronyms and definition of acronyms. There are specific instances where acronyms have not been defined prior to their use e.g. WHO in page 5, line 97 or where an already-described acronym is not used e.g CDF in line 267. Additionally, I would recommend that the authors re-consider the value of including already-published findings from the pre-versus post household surveys. These are not highlighted in the Results section of the Abstract and subsequently, appear to be glossed over.

Reviewer #2: I enjoyed reading the paper and congratulate the authors for linking public health education and community participation.

There are many interesting and important thematic threads in the paper. However, I struggled to discern the primary story the authors aim to tell. The authors specify that they aimed to "test a Community Dialogue Approach (CDA) to enhance schistosomiasis prevention and control" and "evaluate its feasibility, acceptability and potential to improve communities’ level of knowledge, attitudes and practices, and participation in improving schistosomiasis prevention and control". They also state that their intention is to assess how CDA may improve the uptake of health services and promote behaviours recommended by health professionals to reduce and prevent schistosomiasis. The aims are stated differently in diffrent parts of the paper. A reading of the results and discussion raises questions about whether the aims are framed to reflect the contents of the paper.

In my reading the authors seem to be going in different directions. Part of the focus seems to be on evaluating the utility and outcome of the CDA: did it help effect changes in the seletected communities' knowledge, attitudes and practices related to schistosomiasis prevention and control; a part seems to go towards assessing the benefits of using selected community members as non-specialist volunteers; another part seems to be interested in demonstarting how CDA embodies participatory community engagement around a priority health issue in Nampula Province, Mozambique.

These multiple foci don't come together coherently which in part accounts for the long length of the article. The many descriptive details adds to the confusion about the primary aim of the paper and the study reported on.

The authors need to carefully re-consider their focus and formulate firmer and clearer aims and objectives for the paper.

Irrespective of the aims and objectives that the authors settle on, it would be important to review the salient literature on community participation and the use of educational strategies in the public health sector both globally and in Mozambique. This literrature review is missing in the paper.

Aside from describing the many activities that may represent a participatory orientation, the authors need to offer a clear conceptualisation of community participation (CP) as used in their study. It is possible to plot CP on a continuum to show that it may range from mere consulation on one end to deep engagement on the other end. From the descriptions provided in the paper the approach to CP in the study does not seem to approximate a deep engagement. The authors may be critiqued for simply moblising local community change agents to conduct community dialogues intended to create receptivity and acceptance of a public health message formulated by researchers. Community residents serving as community dialogue facilitators seemed to have participated in how best to communicate the public health message on schistosomiasis prevention and control. The researchers seem to have pre-decided that the public health message should focus on the individual behavioural dimensions of schistosomiasis prevention and control.

A more nuanced reading of CP as used in this study would engage with the power dynamics and messiness inherent to CP in health and social led community engagement.

Did participants in the community dialogues point out the social drivers of schistosomiasis and if so how were these addressed in the messaging process? While there is a fleeting reference to Paulo Freire's "critical pedegogy" how was this concept used and what are the kinds of novel insights they helped bring forth? For instance, the authors do not say anything about how community meanings on disease etiology were engaged during the dialogues.

In brief the authors need to offer a nuanced conceptualisation of CP and then locate their work and the use of the CDA within their conceptualisation. How would one evaluate the application of CDA against the key values of CP?

The authors also need to briefly review a few of the relevant debates on the use of educational strategies in public health; what does the evidence tell us about the effects and outcomes of educational strategies? And how do the findings speak to the key issues raised by the literature on educational strategies?

Below is a listing of a few other issues that require the authors attention:

On page 5 the authors refer to a "fragile country". This is an idelogically loaded term. What constitutes a "fragile country"?

On page 6 and elsewhere in the paper the authors use the term "developing country" which also carries problematic meanings. The use of such terms need to be problematised.

If the ideas of 'feasibility' and 'acceptability' are retained in the aims statement they would need to be clearly defined.

As part of the conceptualization of CP the authors would need to clarify what is meant by 'collective decision making'; 'consultation'; 'involvement' and other such terms referring to features of CP.

In the methdology section it would be instructive to review briefly the difficulties and limitations of focus group discussions especially as they relate to the question of participant response set and the dominance of vocal voices. These issues influence participation.

On page 19 the authors state: "discussion guides were not translated into Macua, as there is no tradition of reading or writing in this language...." In the absence of contextual details and information on modes of knowledge making and transmission in the specfic cultural context, such a claim may be intrepreted as problematic. The colonial sciences especially the works of anthropologists and European explorers constructed Africa and its people as deficient and lacking hsitory and knowledge systems. So it would be critical for the authors to provide the neccessary background and details here. Are the authors hinting at an oral tradition of making knowledge?

Lastly, did the researchers do any back translations of the transcripts from English to Macua to ensure optimal accuracy of the translation of texts?

6. PLOS authors have the option to publish the peer review history of their article (what does this mean?). If published, this will include your full peer review and any attached files.

Reviewer #1: Yes: Vernon Mochache

Reviewer #2: No

---

## [Author Response · Author response to Decision Letter 0]

20 Sep 2020

Dear Mr. Enrique Castro-Sánchez,

We thank you and the reviewers very much for a thorough reading and constructive and helpful criticism of our manuscript and for the opportunity to revise and resubmit.

Thank you for the comments of the reviewers and the editors. We found the comments very constructive, and sincerely appreciate the time and attention dedicated to this work. 

We are pleased to submit the improved research article for your consideration.

Below, we have addressed each of these comments and give a point-by-point response. We have numbered the reviewers’ comments and respond using bold font. Note that line numbers in our responses refer to the revised manuscript without tracked changes.

Journal Requirements 

Query 1: Please ensure that your manuscript meets PLOS ONE's style requirements, including those for file naming. 

Thank you for pointing this out. Accordingly, we have reviewed the guidelines and adjusted the manuscript’s style to the journal’s requirements, with special attention to file naming, formatting and citing of figures, tables and supplementary files.

Query 2: Please include the registration number for the clinical trial referenced in the manuscript.

No clinical trial was referenced in the paper. The word ‘trialled’ used at Line 279 refers to ‘other CD models previously trialled and published in Mozambique’, with reference to two studies, none of which were clinical trials. As stated under the Ethics section of the manuscript, for this study, ‘Ethical approval for the study was granted by the University of Leeds School of Medicine Research Ethics Committee (SoMREC/13/071) and the Comité Nacional de Bioética para Saúde in Mozambique (42/CNBS/2014)’. 

Query 3: We note that you have indicated that data from this study are available upon request.

Indeed, as per the consent forms used in this study and approved by the ethics committees referenced in the article, the use of individual participant data for future research purposes is conditional on ethical approval for additional research questions. Data access requests will be reviewed by Malaria Consortium’s Research Group, which can be contacted at research.lead@malariaconsortium.org. Interested researchers do not need membership with Malaria Consortium’s Research Group to gain access to data. The Research Group will review if ethical approval is required and if it has been obtained before making the data available to the researcher.

Equivalent statements have been included in two previous articles relating to this study and published in PLOS journals:

Rassi C, Kajungu D, Martin S, Arroz J, Tallant J, Zegers de Beyl C et al. Have you heard of schistosomiasis? Knowledge, attitudes and practices in Nampula province, Mozambique. PLoS Neglected Tropical Diseases. 2016;10(3): e0004504.

Rassi C, Martin S, Graham K, de Cola MA, Christiansen-Jucht C, Smith LE et al. Knowledge, attitudes and practices with regard to schistosomiasis prevention and control: Two cross-sectional household surveys before and after a Community Dialogue intervention in Nampula province, Mozambique. PLoS Negl Trop Dis. 2019;13(2): e0007138.

Query 4: We note that [Figure 2] in your submission contain map images which may be copyrighted.

A new map has been designed for this publication, and consent to publish has been obtained from the designer (see uploaded file: 2020-09 Content Permission Form Fig 2.pdf)

Reviewer 1

1.1. Comment asking to check the consistent use of acronyms and definition of acronyms. 

This was corrected, specifically on line 94 WHO acronym and line 275 (CDF acronym).

1.2. Comment recommending to re-consider the value of including already-published findings from the pre-versus post household surveys.

Thank you for the suggestion. However, as per Reviewer 2’s request for clarity and focus, supported by the Editor, we have re-focused the paper on the feasibility and acceptability of the CDA. Consequently, we decided not to elaborate further on previously published findings relating the outcomes of the CDA on knowledge, attitudes and practices.

Reviewer 2

2.1. Comment requesting to re-consider the paper’s focus and formulate firmer and clearer aims and objectives.

We agree with the reviewer’s assessment. Accordingly, we have reviewed the scope of the paper to focus on feasibility and acceptability of the CDA in the context of NTD programming. Accordingly, throughout the manuscript, we made the necessary edits where needed to ensure that the respective aims and objectives of the CDA, the pilot study, and this manuscript are clearly spelt out as follow:

The CDA, in the context of schistosomiasis, aims at improving communities’ uptake of recommended prevention and treatment measures, such as MDA adherence, seeking help from qualified health care providers and adopting basic hygiene and sanitation protective practices (intervention description Lines 161 to 164)

The study aimed to test the CDA in the context of schistosomiasis prevention and control, and to evaluate its feasibility, acceptability and potential to improve communities’ level of knowledge, attitudes and practices, with the overall aim of improving schistosomiasis prevention and control (Introduction, lines139 to 142).

This paper presents a detailed description of the CD intervention, and summarise findings with regard to its feasibility and acceptability, drawing on the qualitative and process evaluation data collected throughout the study (Introduction, lines 147 to 149).

The main findings summarized in the conclusion (lines 1270 to 1277) have been re-aligned to speak to the feasibility and acceptability of the intervention.

The revised focus on feasibility and acceptability is reflected in substantial modifications made in the Results and Discussion sections, to ensure that the data presented supports the key findings. 

2.2. Comment asking for definitions of the concepts of 'feasibility' and 'acceptability'.

Definitions have been included in this revised copy in the Methods section, Study design, lines 325 to 338. Feasibility is conceived and assessed along the five dimensions recommended by UK Medical Research Council’s guidance: 1) fidelity, 2) reach, 3)dose delivered, 4) adaptation and 5) mechanisms of impact. We define acceptability as per Peters et al. (2008) which in the context of health interventions, refers to the degree of responsiveness of the intervention to the social and cultural expectations of individuals and communities. 

2.3. Comment asking to review the salient literature on community participation and the use of educational strategies in the public health sector both globally and in Mozambique. 

Thank you for this suggestion. The salient literature on community participation in health programmes was reviewed, including past and recent systematic literature and umbrella reviews (Questa et al. 2020; Draper et al. 2010; Farnsworth et al. 2014; Rosato et al. 2008; Rifkin et al. 2014; Rifkin et al. 2000), and referenced in the paper (Introduction, lines 117to 122).

2.4. Comment requesting for a nuanced conceptualisation of community participation and situating the use of the CDA within this concept. 

We fully agree on the importance of defining and measuring community participation. We wish to clarify that it was beyond the scope of this study to increase community participation in NTD programme. The purpose of testing the CDA was to improve the efficiency and reach of health promotion efforts of the NTD programme. Our assumption was that the use of participatory communication techniques could improve knowledge, attitudes and practices, and contribute to improving communities’ uptake of recommended prevention and treatment measures (as stated in the intervention description section). 

Considering that the CDA could provide a platform for the NTD programme to engage with communities, we agree with the need to situate the CDA within the community participation continuum, and have done so in the Discussion section of the paper at lines 1085 to 1092. We situated the CDA using the modified scale proposed by Draper et al. ranging from mobilisation to collaboration and empowerment. The CDA can be located at the lower level of participation, described as community mobilisation: it involves enabling selected community members to conduct participatory CDs. However, topics and tools were predetermined outside of the community.

We agree that it would have been interesting to explore this aspect in more depth. We attempted to apply Draper et al.’s framework for the assessment of community participation. However, the depth of the data collected (see limitations mentioned at lines 1231 to 1233) was not sufficient to conduct a meaningful analysis of the five dimensions of community participation (needs assessment, leadership, management, organisation and resource mobilisation). Consequently, we decided not to include this dimension in the paper.

2.5. Comments asking for revision of some of the language included in the paper.

Thank you for pointing this out. This was addressed as outlined below.

• ‘The authors refer to a "fragile country". This is an ideologically loaded term’.

This problematic wording has been removed as follows:

• ‘The authors use the term "developing country" which also carries problematic meanings’.

At line 120, the term ’developing country’ was used, it has been replaced by ‘low and middle income countries (LMIC) in this revised copy.

2.6. Comments pointing out unaddressed limitations in the methodology of the study.

• Reviewer: ‘In the methodology section it would be instructive to review briefly the difficulties and limitations of focus group discussions especially as they relate to the question of participant response set and the dominance of vocal voices’.

We have added this aspect as an additional limitation of the study at lines 1247 to 1248.

• Reviewer: ‘On page 19 the authors state: "discussion guides were not translated into Macua, as there is no tradition of reading or writing in this language...." In the absence of contextual details and information on modes of knowledge making and transmission in the specific cultural context, such a claim may be interpreted as problematic’. 

Contextual information has been added at lines 415 to 416. In Mozambique, Portuguese was adopted after colonization as the official language and is the general language of education; only in 2018 local languages were introduced in primary schools . Consequently, adults who are literate are more comfortable reading and writing in Portuguese and often do not read and write easily in their local language.

• Reviewer: ‘Lastly, did the researchers do any back translations of the transcripts from English to Macua to ensure optimal accuracy of the translation of texts?’

As stated at lines 420 to 426, all FGDs were conducted in local language, audio-recorded and subsequently transcribed verbatim, including non-verbal clues, into Portuguese from the audio recording. All the data set, including transcripts, was produced in Portuguese, and not translated into English. Only quotes originally in Portuguese in the transcripts have been translated into English for this publication. These quotes were not back translated into Macua. A sentence was added in the limitations section lines 1235 to 1237 of the manuscript to reflect this potential weakness.

We hope these revisions meet the editor and reviewers’ expectations. On behalf of my co-authors, I thank you for your consideration of this re-submission. We appreciate your time and look forward to your response.

Sincerely yours,

Sandrine Martin

---

## [Decision Letter · Decision Letter 1]

28 Mar 2021

PONE-D-19-34788R1

Engaging affected communities in the prevention and control of schistosomiasis: evaluating the feasibility and acceptability of a Community Dialogue intervention in Nampula province, Mozambique

PLOS ONE

Dear Dr. Martin,

Thank you for submitting your manuscript to PLOS ONE. After careful consideration, we feel that it has merit but does not fully meet PLOS ONE’s publication criteria as it currently stands. Therefore, we invite you to submit a revised version of the manuscript that addresses the points raised during the review process.

We look forward to receiving your revised manuscript.

Kind regards,

David Joseph Diemert, M.D.

Academic Editor

PLOS ONE

Reviewers' comments:

Reviewer's Responses to Questions

**Comments to the Author**

1. If the authors have adequately addressed your comments raised in a previous round of review and you feel that this manuscript is now acceptable for publication, you may indicate that here to bypass the “Comments to the Author” section, enter your conflict of interest statement in the “Confidential to Editor” section, and submit your "Accept" recommendation.

Reviewer #2: All comments have been addressed

Reviewer #3: (No Response)

2. Is the manuscript technically sound, and do the data support the conclusions?

Reviewer #2: Yes

Reviewer #3: Yes

3. Has the statistical analysis been performed appropriately and rigorously? 

Reviewer #2: Yes

Reviewer #3: N/A

4. Have the authors made all data underlying the findings in their manuscript fully available?

Reviewer #2: No

Reviewer #3: Yes

5. Is the manuscript presented in an intelligible fashion and written in standard English?

Reviewer #2: Yes

Reviewer #3: Yes

6. Review Comments to the Author

Reviewer #2: I thank the authors for addressing all the comments and suggestions with care and exactness. I recommend acceptance for publication.

Reviewer #3: This is a very good and important study. However, I have a few concerns that I think if addressed would improve the quality of the paper.

7. PLOS authors have the option to publish the peer review history of their article (what does this mean?). If published, this will include your full peer review and any attached files.

Reviewer #2: No

Reviewer #3: **Yes: **Adam Silumbwe

---

## [Author Response · Author response to Decision Letter 1]

3 May 2021

We found the comments very constructive, and sincerely appreciate the time and attention dedicated to this work. 

We are pleased to submit the improved research article for your consideration.

Below, we have addressed each of these comments and give a point-by-point response. We have numbered the reviewers’ comments and respond using bold font. Note that line numbers in our responses refer to the revised manuscript without tracked changes.

Reviewer 3

1 Title is too long.

The title was edited and shortened.

2 Abstract: Check tense and insert a sentence or two on mechanisms of impact.

The tense was harmonized throughout and a sentence on mechanism of impact was inserted in both the abstract and the author summary.

3 Methods: differentiate two data analysis sections; conceptualize feasibility and acceptability and decide whether to stick with focus on feasibility and acceptability or evaluating implementation process.

In the Methods section, the data analysis sub-headings were edited to specific data analysis of monitoring data and data analysis of qualitative data.

We agree with the reviewer’s advice for clarity and focus of the paper and overall conceptualization of the study. Accordingly, we have revised the scope of the paper to focus on feasibility and acceptability of the CDA in the context of NTD programming. 

Throughout the abstract (lines 29-32), author summary (lines 60-64) and introduction (lines 145-148) sections of the manuscript, we made the necessary edits to ensure that the aims and objectives of study are consistently spelt out.

Definitions of the concepts of 'feasibility' and 'acceptability' have been included in this revised copy in the Methods section, Study design, lines 338 to 353. Feasibility is conceived and assessed along the five dimensions recommended by UK Medical Research Council’s guidance: 1) fidelity, 2) reach, 3) dose delivered, 4) adaptation and 5) mechanisms of impact. We define acceptability as per Peters et al. (2008) which in the context of health interventions, refers to the degree of responsiveness of the intervention to the social and cultural expectations of individuals and communities. 

4 Results:

4.1 Improve the presentation and organisation of results for clarity of the themes and sub-themes 

As per reviewer’s suggestion, a table has been inserted at the beginning of the results section which summarizes the main themes and sub-themes under each component (feasibility and acceptability) of the evaluation; a numbering system of the main themes and sub-themes was also adopted as per reviewer’s suggestion, to help the reader navigate the findings. We hope this has brought clarity.

4.2 Ensure themes and sub-themes headings are self-explanatory

In line with the organization of main themes and sub-themes, respective headings and sub-headings were inserted were missing and edited where needed to ensure they can be understood by reader without prior knowledge of the intervention. 

4.3 Some thematic areas lacked quotes

Relevant quotes were added to the sub-themes of ‘CD attendance’ (lines 675-679) and ‘support to CDFs’ (lines 752-755) which draw on FGD data.

4.4 Reduce repetitions and overlaps across themes

With the re-focusing of the paper on the feasibility and acceptability dimensions of the intervention’s evaluation, and the re-organization of themes and sub-themes under these two components, repetitions and overlap have been addressed.

5 Discussion requires substantial modifications to align with the reorganization of the results.

The revised focus on feasibility and acceptability is reflected in substantial modifications made in the Discussion section, to ensure that the data presented supports the key findings. The main findings summarized in the conclusion (lines 1317 to 1327) have also been re-aligned to speak to the feasibility and acceptability of the intervention.

We hope these revisions meet the editor and reviewers’ expectations. On behalf of my co-authors, I thank you for your consideration of this resubmission. We appreciate your time and look forward to your response.

Sincerely yours,

Sandrine Martin

---

## [Decision Letter · Decision Letter 2]

14 Jun 2021

PONE-D-19-34788R2

Evaluating the feasibility and acceptability of a Community Dialogue intervention in Nampula province, Mozambique

PLOS ONE

Dear Dr. Martin,

Thank you for submitting your manuscript to PLOS ONE. After careful consideration, we feel that it has merit but does not fully meet PLOS ONE’s publication criteria as it currently stands. Therefore, we invite you to submit a revised version of the manuscript that addresses the points raised during the review process.

We look forward to receiving your revised manuscript.

Kind regards,

David Joseph Diemert, M.D.

Academic Editor

PLOS ONE

Journal Requirements:

Reviewers' comments:

Reviewer's Responses to Questions

**Comments to the Author**

1. If the authors have adequately addressed your comments raised in a previous round of review and you feel that this manuscript is now acceptable for publication, you may indicate that here to bypass the “Comments to the Author” section, enter your conflict of interest statement in the “Confidential to Editor” section, and submit your "Accept" recommendation.

Reviewer #3: All comments have been addressed

2. Is the manuscript technically sound, and do the data support the conclusions?

Reviewer #3: Yes

3. Has the statistical analysis been performed appropriately and rigorously? 

Reviewer #3: N/A

4. Have the authors made all data underlying the findings in their manuscript fully available?

Reviewer #3: Yes

5. Is the manuscript presented in an intelligible fashion and written in standard English?

Reviewer #3: Yes

6. Review Comments to the Author

Reviewer #3: Thank for responding to the comments I raised in the earlier version. I notice an improvement in the quality of the manuscript based on the revisions provided. Generally, most of the comments have been addressed. A minor comment is on the first sub-theme sensitization under fidelity. The narrative description does not seem to clearly speak about fidelity. I would expect perhaps, a description of how sensitization(demand creation) was done and what the evaluation found if this was done to plan, in addition a quote to support. In the result and discussion: sub-headings should be complete: Feasibility of using the community dialogue in the prevention and control of schistosomiasis, and Acceptability of the community dialogue in the prevention and control of schistosomiasis. Avoiding headings saying acceptability and feasibility (of what?). The section on limitations, should instead read strengths and limitations.

7. PLOS authors have the option to publish the peer review history of their article (what does this mean?). If published, this will include your full peer review and any attached files.

Reviewer #3: **Yes: **Adam Silumbwe

---

## [Author Response · Author response to Decision Letter 2]

27 Jun 2021

Evaluating the feasibility and acceptability of a Community Dialogue intervention in the prevention and control of schistosomiasis in Nampula province, Mozambique

Sandrine Martin, Christian Rassi, Valdimar Antonio, Kirstie Graham, Jordana Leitão, Rebecca King, Ercilio Jive

Manuscript submitted for review to PLOS ONE (PONE-D-19-34788)

Authors’ response to reviewers’ comments

Dear Mr. David Joseph Diemert,

We thank you and the reviewer very much for a thorough reading and constructive and helpful criticism of our manuscript and for the opportunity to revise and resubmit.

We found the comments very constructive, and sincerely appreciate the time and attention dedicated to this work. 

We are pleased to submit the improved research article for your consideration.

Below, we have addressed each of these comments and give a point-by-point response. We have numbered the reviewers’ comments and respond using bold font. Note that line numbers in our responses refer to the revised manuscript without tracked changes.

Journal Requirements 

Query 1: Please ensure that your manuscript meets PLOS ONE's style requirements, including those for file naming. 

Thank you for pointing this out. Accordingly, we have reviewed the guidelines and adjusted the manuscript’s style to the journal’s requirements, with special attention to file naming, formatting and citing of figures, tables and supplementary files.

Query 2: Please include the registration number for the clinical trial referenced in the manuscript.

No clinical trial was referenced in the paper. The word ‘trialled’ used at Line 285 refers to ‘other CD models previously trialled and published in Mozambique’, with reference to two studies, none of which were clinical trials. As stated under the Ethics section of the manuscript, for this study, ‘Ethical approval for the study was granted by the University of Leeds School of Medicine Research Ethics Committee (SoMREC/13/071) and the Comité Nacional de Bioética para Saúde in Mozambique (42/CNBS/2014)’. 

Query 3: We note that you have indicated that data from this study are available upon request.

Indeed, as per the consent forms used in this study and approved by the ethics committees referenced in the article, the use of individual participant data for future research purposes is conditional on ethical approval for additional research questions. Data access requests will be reviewed by Malaria Consortium’s Research Group, which can be contacted at research.lead@malariaconsortium.org. Interested researchers do not need membership with Malaria Consortium’s Research Group to gain access to data. The Research Group will review if ethical approval is required and if it has been obtained before making the data available to the researcher.

Equivalent statements have been included in two previous articles relating to this study and published in PLOS journals:

Rassi C, Kajungu D, Martin S, Arroz J, Tallant J, Zegers de Beyl C et al. Have you heard of schistosomiasis? Knowledge, attitudes and practices in Nampula province, Mozambique. PLoS Neglected Tropical Diseases. 2016;10(3): e0004504.

Rassi C, Martin S, Graham K, de Cola MA, Christiansen-Jucht C, Smith LE et al. Knowledge, attitudes and practices with regard to schistosomiasis prevention and control: Two cross-sectional household surveys before and after a Community Dialogue intervention in Nampula province, Mozambique. PLoS Negl Trop Dis. 2019;13(2): e0007138.

Query 4: We note that [Figure 2] in your submission contain map images which may be copyrighted.

A new map has been designed for this publication, and consent to publish has been obtained from the designer (see uploaded file: 2020-09 Content Permission Form Fig 2.pdf)

Reviewer 1

1.1. Comment asking to check the consistent use of acronyms and definition of acronyms. 

This was corrected, specifically on line 100 WHO acronym and line 164 (CDF acronym).

1.2. Comment recommending to re-consider the value of including already-published findings from the pre-versus post household surveys.

Thank you for the suggestion. However, as per Reviewer 2’s request for clarity and focus, supported by the Editor, we have re-focused the paper on the feasibility and acceptability of the CDA. Consequently, we decided not to elaborate further on previously published findings relating the outcomes of the CDA on knowledge, attitudes and practices.

Reviewer 2

2.1. Comment requesting to re-consider the paper’s focus and formulate firmer and clearer aims and objectives.

We agree with the reviewer’s assessment. Accordingly, we have reviewed the scope of the paper to focus on feasibility and acceptability of the CDA in the context of NTD programming. Accordingly, throughout the manuscript, we made the necessary edits where needed to ensure that the respective aims and objectives of the CDA, the pilot study, and this manuscript are clearly spelt out as follow:

The CDA, in the context of schistosomiasis, aims at ‘improving communities’ uptake of recommended prevention and treatment measures, such as MDA adherence, seeking help from qualified health care providers and adopting basic hygiene and sanitation protective practices’. (intervention description Lines 176 to 179)

The study aimed to ‘test the CDA in the context of schistosomiasis prevention and control, and to evaluate its acceptability and feasibility to improve communities’ level of knowledge, attitudes and practices, and engagement in wider schistosomiasis prevention and control efforts’. (Introduction, lines 145 to 148).

This paper presents a ‘detailed description of the CD intervention, and summarise findings with regard to its feasibility and acceptability, drawing on the qualitative and process evaluation data collected throughout the study’ (Introduction, lines 153 to 155).

The main findings summarized in the conclusion (lines 1325 to 1332) have been re-aligned to speak to the feasibility and acceptability of the intervention.

The revised focus on feasibility and acceptability is reflected in substantial modifications made in the Results and Discussion sections, to ensure that the data presented supports the key findings. 

2.2. Comment asking for definitions of the concepts of 'feasibility' and 'acceptability'.

Definitions have been included in this revised copy in the Methods section, Study design, lines 338 to 353. Feasibility is conceived and assessed along the five dimensions recommended by UK Medical Research Council’s guidance: 1) fidelity, 2) reach, 3)dose delivered, 4) adaptation and 5) mechanisms of impact. We define acceptability as per Peters et al. (2008) which in the context of health interventions, refers to the degree of responsiveness of the intervention to the social and cultural expectations of individuals and communities. 

2.3. Comment asking to review the salient literature on community participation and the use of educational strategies in the public health sector both globally and in Mozambique. 

Thank you for this suggestion. The salient literature on community participation in health programmes was reviewed, including past and recent systematic literature and umbrella reviews (Questa et al. 2020; Draper et al. 2010; Farnsworth et al. 2014; Rosato et al. 2008; Rifkin et al. 2014; Rifkin et al. 2000), and referenced in the paper (Introduction, lines 123 to 128).

2.4. Comment requesting for a nuanced conceptualisation of community participation and situating the use of the CDA within this concept. 

We fully agree on the importance of defining and measuring community participation. We wish to clarify that it was beyond the scope of this study to increase community participation in NTD programme. The purpose of testing the CDA was to improve the efficiency and reach of health promotion efforts of the NTD programme. Our assumption was that the use of participatory communication techniques could improve knowledge, attitudes and practices, and contribute to improving communities’ uptake of recommended prevention and treatment measures (as stated in the intervention description section). 

Considering that the CDA could provide a platform for the NTD programme to engage with communities, we agree with the need to situate the CDA within the community participation continuum, and have done so in the Intervention Description section of the paper at lines 167 to 173. We situated the CDA using the modified scale proposed by Draper et al. ranging from mobilisation to collaboration and empowerment. The CDA can be located at the lower level of participation, described as community mobilisation: it involves enabling selected community members to conduct participatory CDs. However, topics and tools were predetermined outside of the community.

We agree that it would have been interesting to explore this aspect in more depth. We attempted to apply Draper et al.’s framework for the assessment of community participation. However, the depth of the data collected (see limitations mentioned at lines 1286 to 1290) was not sufficient to conduct a meaningful analysis of the five dimensions of community participation (needs assessment, leadership, management, organisation and resource mobilisation). Consequently, we decided not to include this dimension in the paper.

2.5. Comments asking for revision of some of the language included in the paper.

Thank you for pointing this out. This was addressed as outlined below.

• ‘The authors refer to a "fragile country". This is an ideologically loaded term’.

This problematic wording has been removed as follows:

• ‘The authors use the term "developing country" which also carries problematic meanings’.

At line 83, the term ’developing country’ was used, it has been replaced by low income countries in this revised copy.

2.6. Comments pointing out unaddressed limitations in the methodology of the study.

• Reviewer: ‘In the methodology section it would be instructive to review briefly the difficulties and limitations of focus group discussions especially as they relate to the question of participant response set and the dominance of vocal voices’.

We have added this aspect as an additional limitation of the study at lines 1311 to 1312.

• Reviewer: ‘On page 19 the authors state: "discussion guides were not translated into Macua, as there is no tradition of reading or writing in this language...." In the absence of contextual details and information on modes of knowledge making and transmission in the specific cultural context, such a claim may be interpreted as problematic’. 

Contextual information has been added at lines 429 to 431. In Mozambique, Portuguese was adopted after colonization as the official language and is the general language of education; only in 2018 local languages were introduced in primary schools . Consequently, adults who are literate are more comfortable reading and writing in Portuguese and often do not read and write easily in their local language.

• Reviewer: ‘Lastly, did the researchers do any back translations of the transcripts from English to Macua to ensure optimal accuracy of the translation of texts?’

As stated at lines 426 to 441, all FGDs were conducted in local language, audio-recorded and subsequently transcribed verbatim, including non-verbal clues, into Portuguese from the audio recording. All the data set, including transcripts, was produced in Portuguese, and not translated into English. Only quotes originally in Portuguese in the transcripts have been translated into English for this publication. These quotes were not back translated into Macua. A sentence was added in the limitations section lines 1299 to 1303 of the manuscript to reflect this potential weakness.

Reviewer 3

1 Title is too long.

The title was edited and shortened from ‘Engaging affected communities in the prevention and control of schistosomiasis: evaluating the feasibility and acceptability of a Community Dialogue intervention in Nampula province, Mozambique’ to ‘Evaluating the feasibility and acceptability of a Community Dialogue intervention in the prevention and control of schistosomiasis in Nampula province, Mozambique’.

2 Abstract: Check tense and insert a sentence or two on mechanisms of impact.

The tense was harmonized throughout and a sentence on mechanism of impact was inserted in both the abstract and the author summary.

3 Methods: differentiate two data analysis sections; conceptualize feasibility and acceptability and decide whether to stick with focus on feasibility and acceptability or evaluating implementation process.

In the Methods section, the data analysis sub-headings were edited to specific data analysis of monitoring data and data analysis of qualitative data.

We agree with the reviewer’s advice for clarity and focus of the paper and overall conceptualization of the study. Accordingly, we have revised the scope of the paper to focus on feasibility and acceptability of the CDA in the context of NTD programming. 

Throughout the abstract (lines 29-32), author summary (lines 60-64) and introduction (lines 145-148) sections of the manuscript, we made the necessary edits to ensure that the aims and objectives of study are consistently spelt out.

Definitions of the concepts of 'feasibility' and 'acceptability' have been included in this revised copy in the Methods section, Study design, lines 338 to 353. Feasibility is conceived and assessed along the five dimensions recommended by UK Medical Research Council’s guidance: 1) fidelity, 2) reach, 3) dose delivered, 4) adaptation and 5) mechanisms of impact. We define acceptability as per Peters et al. (2008) which in the context of health interventions, refers to the degree of responsiveness of the intervention to the social and cultural expectations of individuals and communities. 

4 Results:

4.1 Improve the presentation and organisation of results for clarity of the themes and sub-themes 

The presentation of results was reorganized under five main themes for Feasibility and two main themes for Acceptability, and their respective sub-themes. As per reviewer’s suggestion, a table was inserted at the beginning of the results section which lists the main themes and sub-themes using a numbering system, to help the reader navigate the findings. We hope this has brought clarity.

4.2 Ensure themes and sub-themes headings are self-explanatory

In line with the organization of main themes and sub-themes, respective headings and sub-headings were inserted were missing, numbered and edited where needed to ensure they can be understood by reader without prior knowledge of the intervention. 

Under the feasibility component of the evaluation results, ‘Adaptations’ was renamed ’local adaptations of the intervention’ (main theme 1.4); ‘Exploring’ was reworded ‘Exploring the health topic’ (sub-theme 1.5.2); ‘Identifying actions’ was edited as ‘identifying actions to resolve issues’ (sub-theme 1.5.3); ‘Decision making’ was reworded ‘Decision-making on individual and communal actions to be implemented’ (sub-theme 1.5.4).

Under the acceptability component of the evaluation, the themes and sub-themes were carefully worded for clarity.

4.3 Some thematic areas lacked quotes

Relevant quotes were added to the sub-themes of ‘CD attendance’ (lines 681-685) and ‘support to CDFs’ (lines 758-760) which draw on FGD data.

4.4 Reduce repetitions and overlaps across themes

With the re-focusing of the paper on the feasibility and acceptability dimensions of the intervention’s evaluation, and the re-organization of themes and sub-themes under these two components, repetitions and overlap have been addressed.

5 Discussion requires substantial modifications to align with the reorganization of the results.

The revised focus on feasibility and acceptability is reflected in substantial modifications made in the Discussion section, to ensure that the data presented supports the key findings. The discussion was reorganized under three sub-headings to summarize findings and their implications: Feasibility, Acceptability, Embedding CD into disease control programmes.

The main findings summarized in the conclusion (lines 1325 to 1332) have also been re-aligned to speak to the feasibility and acceptability of the intervention.

6. Review Comments to the Author

As per reviewer’s advice, the main themes were reworded ‘Feasibility of using the community dialogue in the prevention and control of schistosomiasis’, and ‘Acceptability of the community dialogue in the prevention and control of schistosomiasis’ in both the results and discussion sections. 

As advised by reviewer, the section on limitations was reworded ‘strengths and limitations’.

As per reviewer’s suggestion, a sentence was added in the sub-theme 1.1.1 Sensitisation to describe how sensitization was done and its fidelity level to the original plan. We could not however insert a relevant quote as the main purpose of this section is rather to explain the lack of fidelity to the original design in the definition of target communities.

We hope these revisions meet the editor and reviewers’ expectations. On behalf of my co-authors, I thank you for your consideration of this resubmission. We appreciate your time and look forward to your response.

Sincerely yours,

Sandrine Martin

---

## [Editor Report · Decision Letter 3]

22 Jul 2021

Evaluating the feasibility and acceptability of a Community Dialogue intervention in the prevention and control of schistosomiasis   in Nampula province, Mozambique

PONE-D-19-34788R3

Dear Dr. Martin,

We’re pleased to inform you that your manuscript has been judged scientifically suitable for publication and will be formally accepted for publication once it meets all outstanding technical requirements.

Kind regards,

David Joseph Diemert, M.D.

Academic Editor

PLOS ONE
---

## [Editor Report · Acceptance letter]

27 Jul 2021

PONE-D-19-34788R3 

Evaluating the feasibility and acceptability of a Community Dialogue intervention in the prevention and control of schistosomiasis in Nampula province, Mozambique 

Dear Dr. Martin:

I'm pleased to inform you that your manuscript has been deemed suitable for publication in PLOS ONE. Congratulations! Your manuscript is now with our production department. 

Kind regards, 

on behalf of

Dr. David Joseph Diemert 

Academic Editor

PLOS ONE